# A quantum optical study of thresholdless lasing features in high-$\beta$ nitride nanobeam cavities

Stefan T. Jagsch[1], Noelia Vico Triviño[2], Frederik Lohof[3], Gordon Callsen[1,2], Stefan Kalinowski[1], Ian M. Rousseau[2], Roy Barzel[3], Jean-François Carlin[2], Frank Jahnke[3], Raphaël Butté[2], Christopher Gies[3], Axel Hoffmann[1], Nicolas Grandjean[2] & Stephan Reitzenstein[1]

Exploring the limits of spontaneous emission coupling is not only one of the central goals in the development of nanolasers, it is also highly relevant regarding future large-scale photonic integration requiring energy-efficient coherent light sources with a small footprint. Recent studies in this field have triggered a vivid debate on how to prove and interpret lasing in the high-$\beta$ regime. We investigate close-to-ideal spontaneous emission coupling in GaN nanobeam lasers grown on silicon. Such nanobeam cavities allow for efficient funneling of spontaneous emission from the quantum well gain material into the laser mode. By performing a comprehensive optical and quantum-optical characterization, supported by microscopic modeling of the nanolasers, we identify high-$\beta$ lasing at room temperature and show a lasing transition in the absence of a threshold nonlinearity at 156 K. This peculiar characteristic is explained in terms of a temperature and excitation power-dependent interplay between zero-dimensional and two-dimensional gain contributions.

[1] Institute of Solid State Physics, Technische Universität Berlin, D-10623 Berlin, Germany. [2] Institute of Physics, École Polytechnique Fédérale de Lausanne, CH-1015 Lausanne, Switzerland. [3] Institute for Theoretical Physics, University of Bremen, D-28334 Bremen, Germany. Correspondence and requests for materials should be addressed to S.R. (email: stephan.reitzenstein@physik.tu-berlin.de)

The search for the limits of semiconductor lasers has initiated the development of micro- and nanolasers with optimized gain material, capable of confining light to nearly diffraction limited volumes[1–3]. Such lasers feature a very high spontaneous emission coupling factor $\beta$, allowing to approach the limiting case of thresholdless lasing[4–11]. With respect to the realization of high-$\beta$ semiconductor nanolasers, one-dimensional photonic crystal nanobeam cavities are excellent candidates, as their design promises small footprint nanolasers[12] combined with an efficient funneling of spontaneous emission into the lasing mode. Since their proposal in 2008, nanobeam cavities have opened up a fast growing field of research with high potential for energy-efficient silicon-integrated nanophotonics[12–15] and low-power on-chip optical data communication[16]. Electrical integration has been successfully demonstrated[12] and their simple geometry features a close to diffraction limited mode volume ($V \sim (\lambda/2n)^3$) and theoretical quality factors $Q$ exceeding $10^7$[17]. This also leads to exciting opportunities in fundamental research, ranging from cavity quantum electrodynamics effects in the single-emitter regime[18] to optogenetics[19]. Of specific relevance to achieve high-$\beta$ lasing is their cavity-mode non-degeneracy and the large mode separation, which allows $\beta$-factors approaching unity[13,20–22]. Together with the efficient carrier confinement inherent to III-nitrides, this makes them an ideal candidate for studying the peculiarities of high-$\beta$ nanolasers under realistic device conditions (room temperature and ambient atmosphere).

The effort to develop low power consuming, i.e. low threshold, nanoscale lasers usually goes hand in hand with the quest to achieve high-$\beta$ lasing. Interestingly though, high-$\beta$ nanolasers do not exhibit an abrupt, phase transition-like, lasing threshold[6]. Instead, high-$\beta$ lasing entails a gradual change in emission properties, including output intensity, linewidth, and the transition from thermal to coherent emission, over a wide range of excitation powers[6–9,23–26]. High-$\beta$ nanolasers should thus not be approached as conventional lasers, but additionally through statistical properties of the emitted radiation[6,7]. We would like to emphasize that the concept of "thresholdless lasing", associated with $\beta = 1$ and the absence of non-radiative losses[8], does not imply a threshold at zero excitation, a concept developed in an early publication[4]. Instead, the gradual transition towards coherent emission always occurs at finite excitation and is thus visible in excitation-dependent second-order autocorrelation measurements[6,7,9]. In practice, most publications on high-$\beta$ lasers still rely solely on input–output (I–O) characteristics in combination with rate equation fitting[27], in particular when it comes to the study of nanolasers operating at elevated temperatures[13,20–22,28]. Thereby, coherence and statistical properties of the emission, which cannot be captured using rate equation modeling, are neglected[6]. In order to preserve a reliable and practically meaningful definition for a lasing threshold, it was repeatedly proposed to rely on statistical properties of the emitted radiation[6,7,9,23,24]. The second-order autocorrelation function $g^{(2)}(\tau) = \langle I(t)I(t-\tau)\rangle/\langle I(t)\rangle^2$, where $\tau$ is the delay between photon counting events in both arms of a Hanbury-Brown and Twiss (HBT) interferometer, is expected to show an excitation power-dependent transition from thermal emission (ideally $g^{(2)}(0) = 2$) to the Poisson limit with $g^{(2)}(0) = 1$ at the onset of lasing. In experiment, a convolution of the correlation function with the detector response function is measured and the thermal emission statistic can only be resolved if the coherence time approaches the detector resolution. Thus, the associated thermal bunching can typically only be observed in the threshold region, where the coherence time is already long enough[23–25]. Measuring solely $g^{(2)}(0) = 1$ above a potential threshold is certainly not a sufficient proof for lasing, as one might merely not be able to resolve the

thermal bunching. The important observation is the excitation power-dependent transition from thermal to coherent emission. A particular signature of high-$\beta$ lasing in the $g^{(2)}(0)$ trace is a transition from thermal to coherent emission over a wide range of excitation powers and a possible deviation from the expected value of 2 in the thermal regime[7,25].

Herein, we present detailed temperature dependent studies of GaN nanobeam lasers grown on a silicon substrate. The nanolasers are based on a single $In_{0.15}Ga_{0.85}N$ quantum well (QW) as gain material and allow us to demonstrate high-$\beta$ lasing at room temperature using continuous wave (cw) excitation. Second-order (intensity) autocorrelation measurements evidence the onset of lasing via an excitation-power-density dependent transition in emission statistics from thermal bunching towards the Poisson limit, associated with the stimulated emission of photons, i.e. coherent light. Our measurements are complemented with a microscopic laser theory to access simultaneously the I–O characteristic, zero time delay photon autocorrelation function $g^{(2)}(0)$, coherence time of the emission, and the carrier population functions. This combination provides access to the underlying lasing physics, in particular to the "ideal" autocorrelation function that is not detector limited. By combining the calculated coherence times with the detector resolution we can simulate the measured autocorrelation function in excellent quantitative agreement with our experimental data. By investigating the power dependence of the photon statistics we are able to observe a lasing transition, even in the absence of a threshold nonlinearity, at a temperature of 156 K. We explore and explain this peculiarity by specific impacts of the gain-dimensionality on the lasing characteristics in a large range of temperatures from 20 K up to room temperature. Of particular interest is a transition region at about 160 K, where additional gain contributions from localized states in the QW lead to a thresholdless I–O characteristic, even for $\beta < 1$. Our results provide a comprehensive analysis of high-$\beta$ lasing and possible pitfalls in its interpretation.

## Results

**Nanolaser design and fabrication.** We address high-$\beta$ lasing at elevated temperatures with nanolasers that are composed of an AlN buffer layer and a 3-nm-thick $In_{0.15}Ga_{0.85}N$ QW embedded in a GaN matrix, which was grown on an Si (111) substrate using metalorganic vapor phase epitaxy. Based on this heterostructure, freestanding nanobeam cavities were subsequently processed by means of e-beam lithography and dry etching techniques (see ref. [13] for more details). The nanobeams, as shown in Fig. 1a, comprise a photonic crystal mirror on both sides, surrounding a taper region with decreasing hole size towards the cavity center, providing a gentle mode confinement in order to increase $Q$[14]. In Fig. 1b the intensity profile of the fundamental mode is plotted, as calculated using a three-dimensional finite difference time domain (3D-FDTD) solver, showing that the mode is well confined to the cavity region with a mode volume $V = 0.63(\lambda/n)^3$. Figs. 1c, d are top and side-view scanning electron microscope images of a typical nanobeam cavity.

**Optical characterization at room temperature.** Optical and quantum-optical characterization of the nanobeam cavities were performed under cw excitation at $\lambda = 325$ nm using a 1/7 chopper wheel to reduce the thermal load. Details on the device fabrication and on the experimental setup are given in the Methods section. The following evaluation is accompanied by the results of a microscopic laser theory for the interaction between the two-dimensional QW gain material and the fundamental cavity mode. Coupled equations for quantum-mechanical expectation values are solved, describing the wave vector-dependent electron

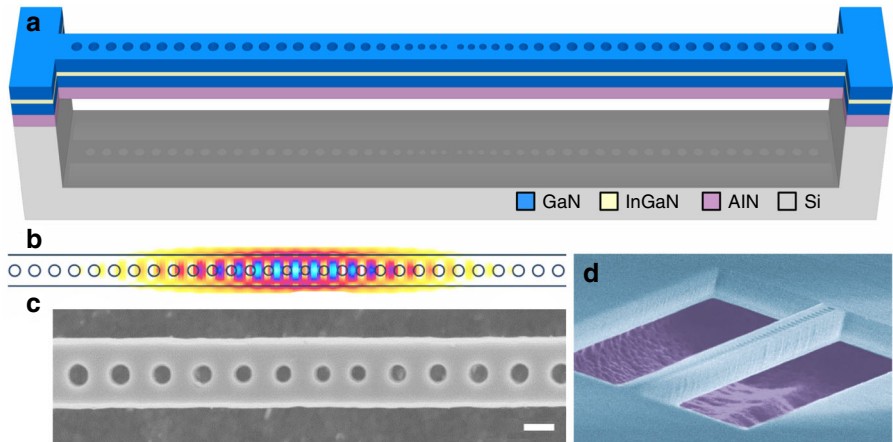

**Fig. 1** III-nitride nanobeam cavity membrane. **a** Schematic drawing of a freestanding nanobeam structure featuring a single InGaN/GaN QW. It consists of two photonic crystal mirrors tapered down to the cavity (not drawn to scale for clarity). **b** Field intensity profile $|E_y|^2$ of the fundamental cavity mode as obtained via 3D-FDTD simulations. **c**, **d** Top and side-view scanning electron microscope images of typical nanobeam structures, where the III-nitride layer, the airgap, and the silicon underneath (false-color in **d**) are noticeable. The scale bar in **c** is equal to 100 nm in length

and hole populations, the intracavity photon number, as well as correlation functions that connect carrier and photonic degrees of freedom. Including these correlation functions provides direct access to the zero time delay photon autocorrelation function $g^{(2)}(0)$, a quantity that is inaccessible in a rate-equation based analysis. Calculating the first-order coherence function $g^{(1)}(\tau)$—and with it the coherence time —allows us to model the decay of $g^{(2)}(\tau)$ from its zero-time-delay value in order to simulate the detector-limited time resolution of the experiment. We would like to emphasize that excellent simultaneous agreement with all experimental data (I–O and $g^{(2)}(0)$ of laser and reference structure) is achieved based on a single set of parameters that enter the model (cf. Supplementary Table 1). Further details of the optical characterization are provided in the methods summary and the theoretical model is described in detail in Supplementary Note 1.

In Fig. 2 we compare room temperature characteristics representative of a lasing and a non-lasing nanobeam cavity, which serves as reference. The main difference between both structures lies in the Q-factor, which has been determined to Q~2200 for the nanobeam laser and Q~1800 for the reference nanobeam. The room temperature I–O curve of the nanobeam laser is depicted in Fig. 2a, together with the cavity mode below threshold (inset). Optical images of the emission are shown in Supplementary Fig. 8. The solid line in Fig. 2a is obtained from the microscopic model, assuming a non-radiative loss rate $A_{nr} = 5 \times 10^7 \, \text{s}^{-1}$, and exhibits a slight threshold nonlinearity before converging to a slope of 1 (dashed line). Due to the strong guiding of photons into the lasing mode, inherent to the nanobeam geometry, emission into non-lasing modes is largely suppressed and the efficient coupling of spontaneous emission into the cavity mode reflected in a $\beta$-factor of ~0.7 for both the lasing and the non-lasing nanobeam . Here, the β-factor can be estimated within the scope of the microscopic model along the lines of [10] taking into account the light-matter coupling strength, as well as radiative losses (cf. Supplementary Table 1). When compared to a rate-equation analysis, the ratio of spontaneous emission into the lasing mode is calculated directly, meaning that the $\beta$-factor is no longer an input parameter (fit parameter) to the theory (see Supplementary Note 1). Nonetheless, the obtained value is in good agreement with the results of a rate equation analysis of the nanobeam laser ($\beta_{RE} = 0.7\pm0.2$) using model and parameters employed in ref. [13]. We conclude that the nonlinear behavior in the I–O curve (Fig. 2a) is mainly caused by non-radiative losses[8]. In comparison, the reference nanobeam has an I–O curve with a

slightly steeper slope of ~1.5 before converging to 1 and finally saturating (Fig. 2d) before reaching the lasing threshold. The absence of lasing is attributed to slightly higher cavity losses (lower Q-factor) and confirmed by autocorrelation measurements and simulation. The I–O curve for the reference nanobeam is well reproduced by theory, accounting for the lower Q-factor of 1800, when compared to the lasing case with Q = 2200. Due to the waveguide nature of the cavity, the mode emission in the vertical direction is monitored via scattered light, unless the far-field emission is further optimized using a sidewall Bragg cross-grating outcoupler[29].

It is important to note that soft nonlinearities in the I–O curve, similar to that observed in Fig. 2a, can in principle also be related to trap filling[30] and thus cannot give an unambiguous proof of lasing. In fact, the observation of further indications of stimulated emission is required to prove lasing, in particular towards the limit of a "thresholdless" laser with a linear I–O curve. Another conventional signature of the onset of lasing is a decrease in the emission linewidth at half maximum (FWHM) and an associated increase in temporal coherence at the transition from predominantly spontaneous emission to stimulated emission. In this context, a power-dependent linewidth narrowing can also be caused by quenching of absorption losses, which complicates the correct interpretation of this lasing indicator in high-$\beta$ lasers. In contrast to conventional lasers (for which $\beta \ll 1$) like, e.g. vertical cavity surface emitting or ridge waveguide lasers, where a pronounced linewidth reduction is an established lasing criterion, high-$\beta$ nanolasers typically show only minor changes in linewidth at threshold[2,3,23,27], and usually deviate from the modified Schawlow-Townes formula for semiconductor-based lasers[31]. Through the soft onset of lasing, which takes place at low carrier densities and with few intracavity photons[6], refractive index fluctuations (gain-refractive index coupling) persist over a large range of excitation powers around threshold and can lead to an almost constant linewidth in the threshold region and beyond[26]. Depending on the heat transport properties of the underlying design, heating of the cavity can also impact the linewidth under high excitation. Considering the discussion above, a lasing threshold could be falsely identified from these classical indicators[32].

In the present case, we observe a minor decrease in the emission linewidth at $P \approx 5 \, \text{kW cm}^{-2}$, superimposed on an overall increasing linewidth (cf. Fig 2b). Linewidth and emission wavelength of the reference cavity (cf. Fig 2e) show a similar

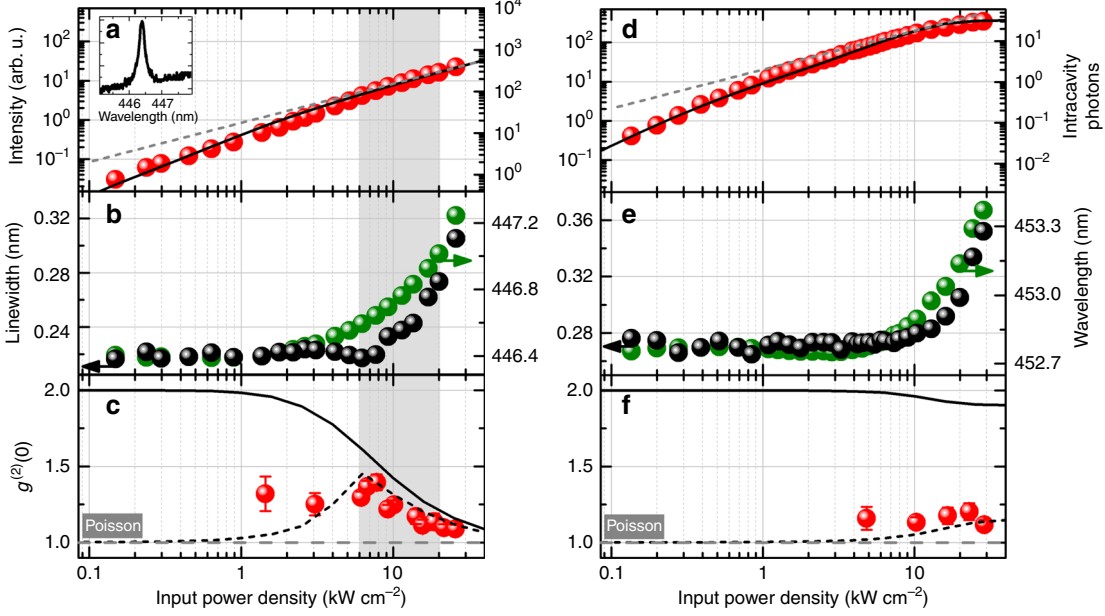

**Fig. 2** Room temperature optical and quantum-optical characterization of a lasing and a non-lasing III-nitride nanobeam cavity. **a**, **d** Room temperature I–O curves. The theoretical model (solid line) in **a** shows a slight nonlinearity before converging to a slope of 1 (indicated by the dashed line). Inset in **a**: Fundamental cavity mode at 0.64 kW cm$^{-2}$. The I–O characteristic in **d** is governed by non-radiative losses and does not show an s-bend before saturating. The increased output intensity with respect to **a** is indicative of an increase in light scattering towards the vertical direction. Note that the intracavity photon number is higher for the lasing nanobeam, which allows building up a coherent photon population. **b**, **e** Resonance peak wavelength (green) and linewidth (FWHM, black). Above about 10 kW cm$^{-2}$ the development of resonance wavelength and emission linewidth is dominated by heating of the cavity region. The lasing structure in **b** exhibits a slight decrease in linewidth around $P \approx 5$ kW cm$^{-2}$. **c**, **f** Second-order autocorrelation function at zero time delay as obtained from experiment (data points) and theory. Proof of the transition to coherent emission (shaded excitation range) is provided by the power dependence of the deconvolved second-order autocorrelation data, showing a clear trend towards the Poisson limit ($g^{(2)}(0) = 1$) with increasing excitation power density. In contrast, the power dependence in **f** reveals a constant $g^{(2)}(0) \leq 1.2$. The evolution of the photon statistics is well reproduced by the microscopic theory (ideal: solid line, convolved: dashed line), when taking into account the calculated coherence time and the convolution with the temporal resolution (~225 ps) of the HBT setup. Error bars in **c** and **f** indicate the standard deviation obtained from fitting the recorded histograms that mirror $g^{(2)}(\tau)$

trend, although a decreasing linewidth at intermediate excitation-power densities is not observed. The simultaneous start of the resonance redshift and the increase in linewidth suggest a thermal origin. Heating of the freestanding nanobeam membrane occurs at higher excitation powers, the resulting thermal expansion of the nanobeam leading to a redshift of the resonance wavelength with increasing excitation power density (cf. Fig 2b, e). Temperature induced fluctuations in the cavity-mode position on timescales faster than the minimum integration time accessible in experiment (10 ms) can result in additional broadening in photoluminescence due to spectral jitter. Similar observations to our experiment were made in refs.[20,22]. Excitation-power-density dependent Raman thermometry measurements and accompanying thermal transport simulations confirm a temperature increase in the cavity region by more than 50 K in the high excitation range during our room temperature experiments (cf. Supplementary Figs. 3 and 4 and Supplementary Note 2). The thermal properties of the nanobeams could be improved, however at the cost of reducing $Q$ and $\beta$, by coupling the cavity region directly to the substrate, using, e.g. the design adopted in ref.[12], where a nanopillar under the cavity is used for electrical injection. As the conventionally sought-after lasing signatures (a pronounced I–O nonlinearity and linewidth decrease) are far more elusive, unambiguous proof of the onset of stimulated emission in high-$\beta$ nanolasers requires excitation power-dependent second-order autocorrelation measurements in order to monitor the change in emission statistics.

**Quantum optical characterization at room temperature**. The results of an excitation power-dependent investigation of the photon statistics are shown in Fig. 2c, f. In order to obtain the zero time delay value $g^{(2)}(0)$, the measured autocorrelation traces were fitted using a convolution of the idealized fitting function and the detector response, taking into account the temporal resolution $\Delta t_{\mathrm{res}} \approx 225$ ps of the HBT setup. See also Supplementary Fig. 5 and Supplementary Note 3 for further details on the fitting procedure. For the lasing nanobeam, we observe clear bunching behavior ($g^{(2)}(0) > 1$), which becomes less pronounced with increasing excitation power density, indicating the transition from spontaneous to dominantly stimulated emission of light (Fig. 2c). The deduction of a high $\beta$-value is supported by a smeared out and incomplete transition to the Poisson limit within the available excitation power density range[25]. While the $g^{(2)}(0)$ signature of a fully Poissonian photon statistic is not observed in experiment, theory suggests that $g^{(2)}(0)$ approaches 1 at ~100 kW cm$^{-2}$. Operation at such excitation power densities would require improved thermal properties of the nanobeam cavities in order to reduce sample heating. The experimental data show a bunching maximum of $g^{(2)}(0) \approx 1.4$ in the threshold region and $g^{(2)}(0) \approx 1.1$ at high excitation. In contrast, in case of the non-lasing cavity, $g^{(2)}(0)$ remains constant at a value of ~1.2 over the investigated range of excitation power densities (Fig. 2f). A thermal influence on the photon statistics via a reduced coherence time is not observed. For the nanobeam laser, theory predicts a clear transition from thermal emission to lasing (solid line in Fig. 2c). The transition is accompanied by an increase in coherence time from ~1 ps to

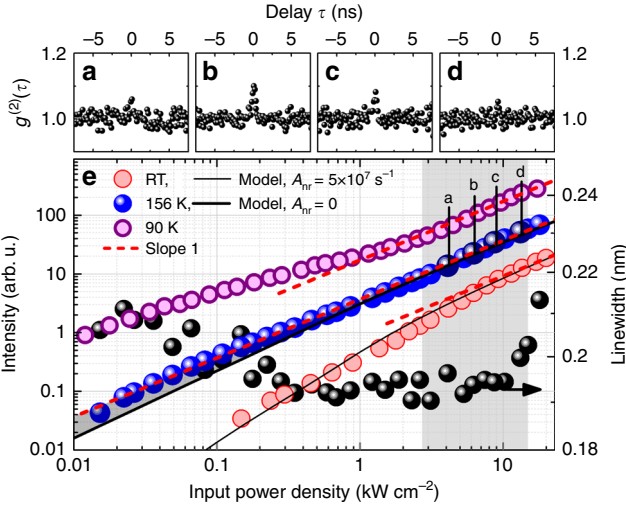

**Fig. 3** Optical and quantum-optical characterization of the thresholdless nanobeam emission at a temperature of 156 K. **a–d** Autocorrelation traces taken at the excitation power densities marked in **e** display the characteristic bunching around threshold (**b**, **c**), which vanishes again for higher excitation (**d**), indicating a transition towards Poissonian emission statistics in the lasing regime (shaded excitation range in **e**). The bunching in **a** cannot be fully resolved due to resolution limitation. **e** Excitation power-dependent I–O curve measured at 156 K (blue) exhibiting a thresholdless behavior and corresponding emission linewidth (black dots). I–O curves measured at room temperature (red) and 90 K (purple) show the development of the I–O characteristics with temperature. Results of our microscopic model, based on a purely 2D QW gain, are shown for RT and 156 K data (solid lines). The I–O curve at 156 K is shown under the assumption of negligible non-radiative losses (nonlinearity solely due to $\beta$ < 1). The increased output intensity in the low excitation regime, due to contributions from localized states, is apparent at 156 K (shaded). Second-order autocorrelation measurements were performed at 156 K at the data points marked a–d. A slope of 1 is indicated by the red dashed lines. The vertical offset was adjusted for clarity

~800 ps across the transition region (cf. Supplementary Fig. 1). Since the decay of the autocorrelation function $g^{(2)}(\tau)$ with respect to $\tau$ is related to the coherence time, the zero-time-delay value can only be resolved if the coherence time exceeds the detector resolution. This becomes apparent in the convolved $g^{(2)}(0)$ trace (dashed line in Fig. 2c), which excellently reproduces the experimental results over the excitation range. For the reference nanobeam (Fig. 2f), the experimentally observed constant value of ~1.2 is traced back to an only moderately increasing coherence time, combined with a calculated $g^{(2)}(0)$ that stays largely thermal until saturation is reached. Furthermore, we find spectral hole burning at the cavity-mode energy in the calculated non-equilibrium carrier distribution functions (Supplementary Fig. 2a), which is another indicator for lasing operation[10].

Finally, we point out that a mean intracavity photon number of one is not necessarily a signature of lasing[6,32]. The theoretical results for both the laser and the reference nanobeam exhibit photon numbers above one, but by comparing Fig. 2a, c, one can infer that a coherent photon population builds up only above a mean photon number of 100 (ref.[32]). It is the imbalance of spontaneous towards stimulated emission that makes the emission coherent above threshold, and this imbalance is not only determined by the mean intracavity photon number, but also by the inversion of the system that is expressed by the non-equilibrium distribution functions for electrons and holes.

## Discussion

In the following, we address the importance of temperature dependent studies to fully explore and correctly interpret the output characteristics of high-$\beta$ nanolasers. In particular, we analyze and discuss the temperature dependence of carrier confinement and non-radiative recombination and their impact on the I–O characteristics. Upon decreasing the sample temperature, we effectively quench non-radiative loss channels so that the kink in the s-shaped I–O curve becomes less pronounced[8]. Depending on the weight of radiative and non-radiative channels, either a soft s-shape or a more pronounced kink in the I–O curve can be observed and modeled for the same $\beta$-factor[8]. As expected, we observe a reduced nonlinearity with decreasing temperature, until approximately 160 K. Below 160 K the I–O curve changes from the familiar s-shape towards an inverse s-shape at low temperatures, exhibiting a completely linear behavior around 156 K (cf. Fig 3e). This thresholdless I–O curve can be observed despite an extracted $\beta$-factor below 1. We model the I–O characteristics at 156 K assuming radiative losses are the same as at RT and non-radiative losses no longer play a significant role ($A_{nr} = 0$). The remaining nonlinearity in the modeled I–O curve originates from the small amount of radiative losses associated with the high, but non-unity, $\beta$-factor.

Instead of ideal spontaneous emission coupling, the thresholdless appearance of the measured I–O curve at 156 K can be ascribed to additional gain contributions from weakly localized (0D) states in the InGaN QW[33,34]. These are activated below a localization temperature of ~ 160 K, as obtained from an Arrhenius evaluation of the QW emission (cf. Supplementary Figs. 6 and 7 and the related discussion in Supplementary Note 4). The result is a two-component 0D–2D gain material. With falling temperature, thermal escape becomes less likely and the number of available localized states increases, leading to increased 0D gain contributions in the low excitation regime (until the localized states are saturated), eventually resulting in an inversely s-shaped I–O characteristic at low temperatures. Around 156 K, these additional gain contributions exactly compensate the threshold nonlinearity and explain the discrepancies between the I–O data and the temperature-dependent modeling under the assumption of a purely 2D gain material (cf. Fig 3e).

As for the room temperature case, the emission linewidth associated with the thresholdless I–O curve at 156 K does not show any pronounced narrowing, which could indicate the transition from spontaneous to stimulated emission. Obviously, a threshold can no longer be identified from the sole I–O characteristics. Thus, Fig. 3e is an excellent example demonstrating that a quantum-optical characterization is required to prove possible lasing in high-$\beta$ nanolasers. We performed such measurements for four excitation power densities, as marked in Fig. 3e. Here, clear bunching at zero time delay appears in the autocorrelation trace and vanishes as the excitation power is further increased (cf. Fig 3a–d). This unambiguously proves the transition in photon statistics at the onset of lasing and highlights the importance of a quantum-optical investigation to demonstrate lasing in the $\beta \to 1$ regime. In comparison to the room temperature measurements, we can only observe the far end of the bunching regime due to resolution limitation (cf. Fig 3a). The visibility of the bunching in $g^{(2)}(\tau)$ is also influenced by the characteristic timescale of intensity fluctuations in the few photon regime (low excitation data points in Fig. 2c, f)[6,7,25], which changes with temperature and is smaller at 156 K in the present system.

In summary, we provide a comprehensive study of high-$\beta$ lasing under realistic device conditions (room temperature and ambient atmosphere) of GaN-based nanobeam cavities on silicon, which is substantiated by microscopic laser theory. By combining

theory and experiment, lasing is unambiguously identified through the simultaneous observation of a transition to coherent emission in the second-order photon correlation function, an increase of the coherence time, spectral hole burning in the carrier population functions, and a slight threshold nonlinearity in the I–O characteristics. Upon decreasing temperature we observe thresholdless lasing at 156 K, which is identified by an excitation-dependent quantum-optical characterization of the emission statistics. We highlight the importance of analyzing the temperature-dependent carrier confinement and the dimensionality of the gain medium to correctly understand and interpret the characteristics of semiconductor nanolasers. We exemplarily show that the different temperature dependence of 0D and 2D gain media crucially impacts the performance of our nanobeam lasers, mimicking thresholdless lasing at a temperature of 156 K despite a $\beta$-factor below 1. Our results give important insights into the manifold peculiarities of semiconductor nanolasers and highlight central issues and pitfalls in the study of high-$\beta$ lasing. To this end, we show that the photon statistics of emission remains a sensitive indicator of a lasing transition, with particular importance in the high-$\beta$ limit.

## Methods

**Experimental setup and measurements**. The 325 nm (3.81 eV) emission line of an HeCd laser was applied for above bandgap cw excitation of the GaN matrix using a 1/7 chopper wheel at 200 Hz in order to reduce the thermal load at high excitation powers. A 20× long working distance UV objective (NA = 0.4) was employed for excitation and emission collection in a combined micro-photoluminescence (μ-PL) and HBT second-order autocorrelation setup. The sample was mounted in a helium flow cryostat. For the μ-PL measurements, the luminescence was dispersed by a single monochromator equipped with a charge-coupled device array. The optical resolution is better than 200 μeV at a photon energy of 2.75 eV. All resulting spectra were calibrated with a mercury gas discharge lamp. For the second-order autocorrelation measurements, the luminescence was guided through a monochromator with an optical resolution of 500 μeV onto a non-polarizing beam splitter and detected by two bialkali photomultiplier tubes in HBT configuration. The measured temporal resolution of the setup is $\Delta t_{res} \approx 225$ ps. Conventional photon counting electronics were used to obtain the final histograms that mirror the second-order autocorrelation function $g^{(2)}(\tau)$ of the nanobeam emission. Room temperature measurements were carried out in a nitrogen atmosphere in order to reduce excitation induced surface depositions over time. All measurements below room temperature were conducted in a controlled low-pressure helium atmosphere in order to avoid excessive sample heating present under vacuum.

**Investigated nanobeam cavities**. More than 1000 nanobeam cavities have been fabricated as part of a processing optimization series (not all cavities are nominally identical), investigating the impact of sample processing on the target parameters based on 3D-FDTD calculations. Approximately 200 nanobeams have then been pre-characterized regarding cavity resonance wavelength, quality factor, and overall output intensity. Subsequently, I–O characteristics have been recorded for 10 nanobeams in the sample region that proved to be the most promising after pre-characterization. Of these nanobeam cavities, five showed indications of a lasing transition in the I–O characteristics. Room temperature power-dependent second-order autocorrelation measurements have been carried out for two lasing and one non-lasing (reference) nanobeam.

**Data availability**. The data supporting the findings presented in this study are available from the corresponding author upon request.

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

## Acknowledgements

The authors acknowledge the contributions of Irene Sánchez-Arribas for the optical characterization of the unprocessed sample. The research leading to these results has received funding from the German Research Foundation via the D-A-CH project Re2974/8-1, and projects Re2974/10-1, Gi1121/1-1, the European Research Council under the European Union's Seventh Framework ERC Grant Agreement No. 615613, the Swiss National Science Foundation Grants Nos 200020-113542, 200020-150202, and 200020-162657 as well as the DACH-FNS Grant 200021E-154668.

## Author contributions

N.G., A.H. and S.R. initiated the research and conceived the experiments. S.T.J., G.C. and S.K. performed the experiments. S.T.J. and S.R. performed the data analysis. J.-F.C. grew the samples which were subsequently processed by N.V.T. following extensive 3D-FDTD simulations. I.M.R. and R. Butté. performed the optical analysis of the unprocessed sample and wrote the corresponding SM sections. G.C. and S.T.J. performed the Raman thermometry measurements and I.M.R performed the COMSOL modeling. F.L., R. Barzel, F.J. and C.G. developed the microscopic model. F.L. and R. Barzel performed the simulations. S.T.J., C.G. and S.R. wrote the manuscript with contributions from all other authors.

## Additional information

**Competing interests:** The authors declare no competing financial interests.

