## [Peer Review File · Nature Communications]

Reviewers' comments:

Reviewer #1 (Remarks to the Author):

In this paper, the authors reported operation of a threshold-less nano-lasers using GaN quantum wells deposited on silicon nitride. While such threshold-less nano-lasing is of scientific interest, the authors' claim this being the central goal of nano-laser community is not only exaggerated, but also misleading. The authors are correct that reduction of threshold for a nano-laser has been of interest in the nano-laser community, however, the primary reason for this is technological, so that the laser can be turned on and off with low power. Such modulation is not efficient with a laser, which does not show nonlinear input-output behavior. Hence, even though, the nonlinear Pin-Pout behavior is not an indication of the lasing, such nonlinear behavior is of utmost importance, as that is when we can have large on-off ratio of laser-modulation, important for optical interconnects. Without such nonlinear behavior, the nano-lasers have little utility, and are not interesting to a broad community of researchers working in nano-lasers.

If this was the only problem of the present work, I would have been fine. However, I am afraid, the paper does not show the conclusive results, the authors were hoping to obtain. Namely, they did not see any $g_2(0)=1$ at zero threshold. In fact, for any photoluminescence measurement, when the system is optically pumped with an above-band laser, at high power the second-order autocorrelation often goes to unity. This does not mean it is a laser. While coherence function of $g_2(0)=1$ is indeed a signature of the lasing, it is by no means a sufficient condition. From the data presented in the paper, it looks like an ordinary power-dependent PL. The data presented on linewidth narrowing is also not convincing. Finally the cavity quality factor is only 2000, which is not good enough to observe threshold-less lasing.

Based on this, I cannot recommend this paper to be published. In the recent past there are several nano-lasers papers. The authors pointed out some of them in the references 22-23. There are similar results reported in other papers, including,
Nature 520, 69-72 (02 April 2015)
Nano Lett., 2015, 15 (8), pp 5302-5306
Nature 461, 629-632 (1 October 2009)

The authors rightly pointed out that none of these papers has conclusively proved the presence of lasing. Unfortunately, neither did the authors. The presence of lasing implies the change in coherence, while signature of the nonlinearity and linewidth narrowing. A true threshold-less laser will not show a nonlinear change, but at that case, the coherence function will be unity at all the power. Hence, I cannot recommend this paper for publication.

Reviewer #2 (Remarks to the Author):

This is an interesting paper on the observation of high-beta laser operation in GaN nanobeam lasers. The results appear to be of a high technical quality. However, I have several comments and questions for the paper:

1) In the Abstract, the authors promote the results as "close to ideal lasing behavior....". However, at the same time the authors show that the lasers have relatively poor thermal properties, even in the case considered, where the pump beam is chopped to reduce the thermal load. What is meant by close to ideal behavior, in particular compared to previous work? What are the prospects of improving the properties - and what is the importance of this work in relation to that?

2) [Related to the above] The authors quote a value of 0.7 for the beta factor. What is the uncertainty of that value? In relation to applications, why is a laser with a beta factor of, say, 0.7, better than one with a beta of, say, 0.3? The authors seem to identify the high value of beta with a

good laser.

3) The authors state that the linewidth shows a small decrease at the onset of lasing. But if this onset is taken to coincide with the reduction of $g(2)$, the reduction seems to happen before the threshold (and actually following a small increase). Anyway this change is small compared to the large increase of the linewidth at higher pump powers. The authors attribute this to thermal effects, but the mechanism is not clear and should be explained.

4) Why doesn't $g(2)$ achieve the value of 2 well below threshold? Is it due to temporal resolution issues, connected with a short coherence time? Is the dependence on the pump power, as eg. displayed in the upper panel of Fig. 4, understood in any quantitative manner?

5) The authors ascribe the thresholdless behavior to a complex 0D-2D gain. However, that paragraph (just below Fig. 4) is very difficult to understand, and I didn't get the point. I think the authors need to make this more quantitative; maybe this is the novel contribution of the paper?

With these comments in mind, I do not think that the authors have demonstrated the novelty or progress and clarity in presentation that is required for publication in Nature Communications.

We thank the reviewers for their detailed evaluation of our manuscript and for their valuable comments and suggestions. We have considered their reports and changed the manuscript accordingly. In the following, we address their comments in a point-to-point response. While the whole manuscript has been subject to improvements according to the reviewers' suggestions, only major changes in direct relation to the questions/remarks are explicitly quoted in the point-to-point response. The reference numbering in the manuscript has changed from [15] onwards.

Reviewer #1 (Remarks to the Author):

Reviewer:

In this paper, the authors reported operation of a threshold-less nano-lasers using GaN quantum wells deposited on silicon nitride. While such threshold-less nano-lasing is of scientific interest, the authors' claim this being the central goal of nano-laser community is not only exaggerated, but also misleading. The authors are correct that reduction of threshold for a nano-laser has been of interest in the nano-laser community, however, the primary reason for this is technological, so that the laser can be turned on and off with low power. Such modulation is not efficient with a laser, which does not show nonlinear input-output behavior. Hence, even though, the nonlinear Pin-Pout behavior is not an indication of the lasing, such nonlinear behavior is of utmost importance, as that is when we can have large on-off ratio of laser-modulation, important for optical interconnects. Without such nonlinear behavior, the nano-lasers have little utility, and are not interesting to a broad community of researchers working in nano-lasers.

Our response:

We thank the reviewer for his/her general comment on our work. In our claim, we had primarily the large and very active scientific community in mind, which deals with fundamental aspects of lasing in micro- and nanocavities, whatever the material system. In this field, a central goal is indeed the realization of thresholdless lasing. Another important goal is lasing based on a single gain center, cf. Ref. 1 entitled "Seeking the Ultimate Nanolaser". Focusing on low-power lasers with large on-off ratio is certainly important for optical interconnects, but for many researcher working on fundamental aspects of nanolasers high β -factors and thresholdless operation is of equal or even higher importance. Moreover, we would like to point out that in contrast to the reviewer's statement about the applicability of high- β nanolasers for optical communication, modulation depth and modulation speed are increased in high- β devices, as has been reported in e.g. [5] section V, PRA 44, 657 (1991) section IV.C and Nature Phys. 2, 484 (2006). The focus review Nature Phot. 8, 908-918 (2014) – added as ref. [16] – on developments in nanolasers also covers photonic crystal based devices and their application in on-chip data communication. See e.g. IEEE Photonics Technol. Lett. 21, 522-524 (2009) for a proof-of-principle demonstration of an on-chip optical interconnect based on another typical high- β nanolaser design (microdisc). In higher order modulation formats, which are in the focus of research concerning increased bandwidth, the performance even hinges on a linear device characteristic in the operation range.

The huge interest in nanolasers is reflected in numerous high impact publications, e.g. [2], [5-6], [8], [23-24]. The topic is also timely important as can be seen by a number of recent publications including those mentioned by the reviewer, e.g. [3], [9-11], [22], [27-28], Nature 520, 69-72 (2015), Nano Lett. 15(8), 5302-5306 (2015) and Nature Mat. 14, 370-371 (2015). Interestingly, in the latter it is explicitly stated that: "[...] A bigger challenge would be the understanding of the fundamental aspects of the laser, such as its coherence properties and photon statistics. This is important to substantiate the true lasing characteristics of nanolasers [...]" – a point that we explicitly address in our manuscript.

We revised the text in order to clarify the motivation and importance of our work in more detail. The relevant paragraph now reads: "*The search for the limits of semiconductor lasers has initiated the development of micro- and nanolasers with optimized gain material and tight light confinement [1-3].*

Such lasers feature very high spontaneous emission coupling factors (β -factors) and allow one to approach the limiting case of thresholdless lasing [4-11]. With respect to the realization of high- β semiconductor nanolasers, 1D photonic crystal nanobeam cavities are very interesting candidates, as their design promises small footprint nanolasers [12] combined with an efficient funneling of spontaneous emission into the lasing mode. Indeed, since their proposal in 2008, nanobeam cavities have opened up a fast growing field of research with high potential, e.g. for silicon integrated nanophotonics [12-15] and low power on-chip optical data communication [16]. Moreover, electrical integrability has recently been successfully demonstrated [12] and their simple geometry features a nearly diffraction limited mode volume ($V \sim (\lambda/2n)^3$) and theoretical quality factors Q exceeding 10^7 [17]. This leads to exciting opportunities in fundamental research, ranging for instance from cavity quantum electrodynamics effects in the single emitter regime [18] to optogenetics [19]. Of specific relevance to achieve high- β lasing is their cavity mode non-degeneracy and the large mode separation, which allows β -factors approaching unity [13], [20-22]. This appealing feature, together with the efficient carrier confinement inherent to III-nitrides, makes them an ideal candidate for studying high- β nanolasers under realistic device conditions (room temperature and ambient atmosphere) and eventually realizing thresholdless lasing at room temperature. Apart from the large fundamental interest in this long-standing goal of the nanolaser and quantum optics community, thresholdless lasing also has important impact from a practical perspective, e.g. concerning the large-scale integration of energy efficient nanolasers in silicon photonics."

Reviewer:

If this was the only problem of the present work, I would have been fine. However, I am afraid, the paper does not show the conclusive results, the authors were hoping to obtain. Namely, they did not see any $g_2(0)=1$ at zero threshold. In fact, for any photoluminescence measurement, when the system is optically pumped with an above-band laser, at high power the second-order autocorrelation often goes to unity. This does not mean it is a laser. While coherence function of $g_2(0)=1$ is indeed a signature of the lasing, it is by no means a sufficient condition. From the data presented in the paper, it looks like an ordinary power-dependent PL. The data presented on linewidth narrowing is also not convincing. Finally the cavity quality factor is only 2000, which is not good enough to observe threshold-less lasing.

Our response:

The identification of lasing via an analysis of the $g^{(2)}(\tau)$ -function has become an established and widely accepted method in the field of cavity enhanced nanolasers. Actually, in order for the second order autocorrelation function to be a valid identifier of a lasing transition, the excitation power dependent transition from thermal to coherent emission statistics (cf. Figs 2 & 3) has to be observed, e.g. [6-7, 9, 23-25]. In the evaluation of corresponding experimental data one needs to take into account the limited temporal resolution of the HBT-setup which leads to a characteristic bunching behavior around threshold and a transition of $g^{(2)}(0)$ towards one above threshold, cf. also [23-25]. In general, in high- β lasers the kink in the I-O curve, the power dependence of the linewidth and the $g^{(2)}(\tau)$ -function are smeared out around threshold, e.g. [6-8], [23-26]. Therefore, one cannot expect $g^{(2)}(0) = 1$ at threshold, but a gradual transition towards 1 above threshold, as we report in our manuscript.

The following part has been added for clarification: "*In experiment, a convolution with the detector response function is measured and the thermal emission statistic can only be resolved if the coherence time exceeds the detector resolution. Bunching can typically only be resolved in the threshold region, where the coherence time is already long enough [23-25]. Measuring solely $g^{(2)}(0) = 1$ above a potential threshold is certainly not a sufficient proof for lasing, as one might merely not be able to resolve the thermal bunching. The important observation is the excitation power dependent transition from thermal to coherent emission, leading to the characteristic bunching behavior around threshold.*"

Regarding the achievability of thresholdless lasing, we would like to point out that the β -factor is determined not only by the Q-factor but also by the mode volume V and the underlying Purcell-factor FP in cQED-enhanced nanolasers. Since FP is proportional to Q/V , a moderate Q can be overcompensated by a small mode volume to achieve a large β -factor. There are also other factors, like the spatial and spectral position of the emitters that influence the β -factor, so that we do not agree that a cavity factor of 2000 should per se not be good enough to observe thresholdless lasing. For instance, Ref. [2] reports a quality factor of $Q \approx 265$ leading to a β -factor of 0.95, as determined from rate-equation fitting (the theoretical value from structural optimization is reported as 0.99). The Q/V ratio for the structures investigated in [2] is indeed similar to ours. Additionally, the measured (loaded) cavity Q is usually lower than the intrinsic quality factor. The measured Q is decreased by loading from the gain medium, as well as possible broadening due to heating, cf. e.g. Armani et al. Nature 421, 925-928 (2003).

Reviewer:

Based on this, I cannot recommend this paper to be published. In the recent past there are several nano-lasers papers. The authors pointed out some of them in the references 22-23. There are similar results reported in other papers, including,

Nature 520, 69-72 (02 April 2015)

Nano Lett., 2015, 15 (8), pp 5302-5306

Nature 461, 629-632 (1 October 2009)

The authors rightly pointed out that none of these papers has conclusively proved the presence of lasing. Unfortunately, neither did the authors. The presence of lasing implies the change in coherence, while signature of the nonlinearity and linewidth narrowing. A true threshold-less laser will not show a nonlinear change, but at that case, the coherence function will be unity at all the power. Hence, I cannot recommend this paper for publication.

Our response:

The papers mentioned by the reviewer clearly highlight the strong scientific interest in low-threshold nanolasers, in particular from a fundamental physics and material science point of view. We would like to stress that none of these papers investigates the power dependence of the $g^{(2)}(\tau)$ -function. In this regard we agree with the reviewer saying that “[...] *none of these papers has conclusively proved the presence of lasing.*” Our comprehensive study of nanobeam lasers, which importantly includes the power dependence of the $g^{(2)}(\tau)$ -function, contains in this sense the first conclusive proof of thresholdless lasing. Moreover, we would like to highlight that we did this at elevated temperatures up to room temperature, while the vast majority (if not all) of the related work on nanolasers has been performed either at cryogenic temperatures and/or under pulsed excitation or lacks an investigation of the photon statistics. When mentioning “*the coherence function will be unity at all the power*” the reviewer seems to expect a value of $g^{(2)}(0)=1$ from zero excitation onwards, i.e. a threshold at zero excitation power. This zero threshold interpretation was developed in an early paper from De Martini et al. (Ref. [4] in the manuscript) and was soon proven false, as e.g. discussed by Rice and Carmichael in Ref. [6]. The power dependent transition in emission statistics from thermal to coherent emission occurs at finite powers even in the limit of $\beta=1$, e.g. Refs. [6-7], [9].

In order to clarify the discussion on the threshold definition as $\beta \rightarrow 1$, we added the following sentence: “*We would like to note that the term “thresholdless” does not imply a threshold at zero excitation, a concept developed in an early publication [4]. Instead, the gradual transition towards coherent emission always occurs at finite excitation and is thus visible in excitation dependent second order autocorrelation measurements [6-7], [9].*”

Reviewer #2 (Remarks to the Author):

Reviewer:

This is an interesting paper on the observation of high-beta laser operation in GaN nanobeam lasers. The results appear to be of a high technical quality. However, I have several comments and questions for the paper:

1) In the Abstract, the authors promote the results as "close to ideal lasing behavior....". However, at the same time the authors show that the lasers have relatively poor thermal properties, even in the case considered, where the pump beam is chopped to reduce the thermal load. What is meant by close to ideal behavior, in particular compared to previous work? What are the prospects of improving the properties - and what is the importance of this work in relation to that?

Our response:

We thank the reviewer for the positive evaluation and for critical reading, helping to improve the manuscript. He/she pointed out correctly that the wording in the abstract could be misleading and was expressed too general. Close-to-ideal lasing refers to the close to ideal coupling of spontaneous emission into the lasing mode, given by the high β factor and resulting in a weakly pronounced threshold nonlinearity in a room temperature quasi-cw measurement.

The sentence in the abstract has been rephrased accordingly: "*We present close-to-ideal spontaneous emission coupling and a nearly linear I-O characteristic in GaN nanobeam lasers grown on silicon.*", aside similar changes throughout the manuscript.

We would like to note that previously comparable results were either obtained for cavities with a larger device footprint, in particular 2D photonic crystal cavities [23], [28], at cryogenic temperatures [2], [23], under pulsed excitation [22], or lack the important investigation of the photon statistics. Our results therefore not only present a substantial step towards realistic device conditions (e.g. room temperature, ambient atmosphere, silicon integration), but also include a detailed quantum-optical characterization of lasing in a high- β device, in particular the first such investigation in the thresholdless regime. The following part has been added to further motivate the use of the nanobeam design:

"Of specific relevance to achieve high- β lasing is their cavity mode non-degeneracy and the large mode separation, which allows β -factors approaching unity [13], [20-22]. This appealing feature, together with the efficient carrier confinement inherent to III-nitrides, makes them an ideal candidate for studying high- β nanolasers under realistic device conditions (room temperature and ambient atmosphere) and eventually realizing thresholdless lasing at room temperature."

Concerning the poor thermal coupling of the nanobeams, the aim of the present design is to achieve maximum possible β values, which can be achieved through a freestanding cavity region. The thermal properties could for example be improved through coupling of the cavity region to the substrate using the design adopted in [12], where the cavity region is connected to the substrate via a nanopillar (in Ref [12] to serve as electrical contact). The following sentence was added to the manuscript to give a prospect of improving thermal properties: "*The thermal properties of the nanobeams could be improved, however at the cost of reducing Q and β , by coupling the cavity region directly to the substrate, using e.g. the design adopted in Ref. [12], where a nanopillar under the cavity is used for electrical injection.*"

Reviewer:

2) [Related to the above] The authors quote a value of 0.7 for the beta factor. What is the uncertainty of that value? In relation to applications, why is a laser with a beta factor of, say, 0.7, better than one with a beta of, say, 0.3? The authors seem to identify the high value of beta with a good laser.

Our response:

The room temperature rate equation fit yields $\beta = 0.7 \pm 0.2$. The error is now stated in the manuscript. A discussion of remaining uncertainties has been added to the corresponding section of the SM, and the main text now reads: “Fitting this curve by a standard rate equation model allows us to determine a β -factor of 0.7 ± 0.2 , in accordance with [13]. We refer to the Supplementary Material (SM) for a detailed discussion of our fitting procedure. This includes a model for the temperature dependence of the employed ABC model, describing defect related Shockley-Read-Hall, radiative and Auger recombination, as well as uncertainties in the extraction of β .” Remaining uncertainties in the extracted β result from uncertainties in the rate equation parameters, mainly from the employed ABC coefficients. Since there are no recombination coefficients available in the literature for InGaN quantum wells grown on silicon substrate, we relied on measurements performed on similar quantum wells grown on c-plane freestanding GaN substrate in order to reduce the degrees of freedom in the rate equation fit.

An additional paragraph has been added to the SM. The paragraph reads: “Remaining uncertainties in β arise through uncertainties in the ABC coefficients taken from [2], which were extracted for InGaN quantum wells grown on c-plane freestanding GaN substrates. Due to a lack of reports on recombination coefficients for InGaN quantum wells grown on silicon substrates the former were adopted in the rate equation fit, leading to $\beta = 0.7 \pm 0.2$.”

Related also to 1), the association between high- β and “a good laser” has been clarified throughout the manuscript and now correctly refers to the increase in β and a reduced threshold nonlinearity. Generally, the device threshold shifts towards lower excitation powers as β increases, cf. e.g. [8]. In relation to applications, a change in β from 0.3 to 0.7, however, should not affect the device performance a lot. The difference is pronounced when comparing a “classical” laser (e.g. VCSEL with typical $\beta \sim 10^{-4}$ or less) to a cavity-QED laser with $\beta > 0.1$. As stated in the manuscript, with increasing β , it becomes harder to even define a threshold (value), as the transition in emission statistics is no longer abrupt.

Reviewer:

3) The authors state that the linewidth shows a small decrease at the onset of lasing. But if this onset is taken to coincide with the reduction of $g(2)$, the reduction seems to happen before the threshold (and actually following a small increase). Anyway this change is small compared to the large increase of the linewidth at higher pump powers. The authors attribute this to thermal effects, but the mechanism is not clear and should be explained.

Our response:

In the limit $\beta \rightarrow 1$ classical lasing indicators, such as linewidth and I-O nonlinearity, no longer coincide with the change in emission statistics. With increasing β there are less carriers and photons in the cavity when the device reaches the transition region and quantum fluctuations play a pronounced role over an increasing region around threshold, suppressing a transition towards coherent emission, cf. [6]. As a result, the transition in $g^{(2)}(0)$ occurs at a different (in particular higher) excitation power. The small increase in linewidth, prior to the decrease around threshold, is in agreement with observations in e.g. [23] and a result of the gain-refractive index coupling, as discussed in e.g. Björk et al., APL 60, 304 (1992).

The discussion of the linewidth has been improved. The paragraph now reads: “One further conventional signature of the onset of lasing is a decrease in the emission linewidth (FWHM) and an associated increase in temporal coherence at the transition from predominantly spontaneous emission to stimulated emission. In this context, it is important to note that also a power dependent linewidth narrowing can be caused by quenching of absorption losses, which complicates the correct interpretation of this lasing indicator in high- β lasers. In contrast to conventional lasers ($\beta \ll 1$), high- β nanolasers typically show only minor changes in linewidth at threshold [2-3], [23], [27]. This is a result of increased refractive index fluctuations (gain-refractive index coupling) in the excitation range around threshold [26]. Through the soft onset of lasing, which takes place already at low carrier densities and with few intracavity photons [6], these fluctuations persist over a large range of excitation powers, leading to an almost constant linewidth in the threshold region [26]. Considering the discussion above, a lasing threshold could be falsely identified from these classical indicators.

In the present case, we observe a minor feature in the emission linewidth, superimposed on a dominantly thermal behavior, at $P \approx 5 \text{ kW/cm}^2$, cf. Fig. 2b. Heating of the freestanding nanobeam membrane occurs at higher excitation powers, with highest temperatures in the cavity region. The resulting thermal expansion of the nanobeam leads to a considerable redshift of the resonance wavelength with increased excitation power density (see also Fig. 2b). The respective impact on the linewidth is twofold. On the one hand, the temperature gradient from the cavity region causes an irregular expansion of the nanobeam membrane and thereby reduces the Q factor, which in turn causes an excitation dependent broadening of the cavity mode. On the other hand, fluctuations in the excitation power lead to shifts in the cavity mode position on timescales faster than the integration time, resulting in an additional broadening in photoluminescence. Similar observations have been made in Refs. [20] and [22]. [...]”

Reviewer:

4) Why doesn't $g^{(2)}$ achieve the value of 2 well below threshold? Is it due to temporal resolution issues, connected with a short coherence time? Is the dependence on the pump power, as eg. displayed in the upper panel of Fig. 4, understood in any quantitative manner?

Our response:

A value below 2 in the thermal regime is expected for high- β devices, as a result of the cutback in non-lasing modes that contribute to noise in the thermal regime, cf. [7] and [25], as well as due to the convolution with the temporal resolution of the detectors. The latter has been accounted for in the employed convolution fit, reducing this particular impact. However, the detector resolution does limit the acquisition of data points at low excitation powers, leading to an artificial $g^{(2)}(0) = 1$ for bunching widths below the resolution limit, e.g. [23-25], also Fig. 3i (former Fig. 4 – see below for the new figure) and Fig. S2 (in the SM) from inspection. The corresponding paragraph now reads:

“In experiment, a convolution with the detector response function is measured and the thermal emission statistic can only be resolved if the coherence time exceeds the detector resolution. Bunching can typically only be resolved in the threshold region, where the coherence time is already long enough [23-25]. Measuring solely $g^{(2)}(0) = 1$ above a potential threshold is certainly not a sufficient proof for lasing, as one might merely not be able to resolve the thermal bunching. The important observation is the excitation power dependent transition from thermal to coherent emission, leading to the characteristic

bunching behavior around threshold. A particular signature of high- β lasing in the $g^{(2)}(0)$ trace is a transition from thermal to coherent emission over a wide range of excitation powers and a deviation from the expected value of 2 in the thermal regime. As β increases, the resulting cutback in non-lasing modes reduces intensity noise already below threshold, resulting in $g^{(2)}(0) < 2$ in the thermal regime [7], [25]."

The discussion of Fig. 3 (previously Fig 4) (upper panels) has been extended to include mention of the temporal resolution limitation of the HBT setup. The following sentence has been added to the figure caption: "*The bunching in i cannot be fully resolved due to resolution limitation.*"

In the main text a reference to Fig. 3i (previously Fig. 4i) has been added: "*In comparison to the room temperature measurements, we can only observe the far end of the bunching regime due to resolution limitation (cf. Fig. 3i).*"

The upper panel of Fig. 3 (previously Fig. 4) should not be understood in a quantitative manner, in the sense that the signal is still convoluted with the detector response and no viable information except for the transition in emission statistics can be extracted from the traces, as they are close to the resolution limit.

Reviewer:

5) The authors ascribe the thresholdless behavior to a complex 0D-2D gain. However, that paragraph (just below Fig. 4) is very difficult to understand, and I didn't get the point. I think the authors need to make this more quantitative; maybe this is the novel contribution of the paper?

Our response:

We agree that the paragraph concerning the evolution of the I-O curve and the influence of the gain material was unclear in the previous form. The manuscript has been modified in order to highlight and better explain this novel observation. Figure 3 (previously Fig. 4) has been changed to better visualize the effect of lowering temperature on the I-O curve and make it easier to follow the amended explanation. The discussion now reads:

"Discussion *In the following, we address the importance of temperature dependent studies to fully explore and correctly interpret the output characteristics of high- β nanolasers. In particular, we analyze and discuss the temperature dependence of carrier confinement and non-radiative recombination and their impact on the I-O characteristics. Upon decreasing the sample temperature, we effectively quench non-radiative loss channels so that the kink in the s-shaped I-O curve becomes less pronounced [8]. Depending on the weight of radiative and non-radiative channels, either a soft s-shape or a more pronounced kink in the I-O curve can be observed and modeled for the same β -factor [8]. As expected, we observe a reduced nonlinearity with decreasing temperature, until approximately 160 K (cf. Fig S1 in the SM). Below 160 K the I-O curve changes from the familiar s-shape towards an inverse s-shape at low temperatures, exhibiting thresholdless behavior around 156 K (cf. Fig. 3a). This thresholdless I-O curve can be observed despite the presence of non-radiative losses and an extracted β -factor below 1.*

Figure 3 | Optical and quantum-optical characterization of the thresholdless nanobeam emission at a temperature of 156 K. **a**, Excitation power dependent I-O curve measured at 156 K (blue) exhibiting a thresholdless behavior and corresponding emission linewidth (black). I-O curves measured at room temperature (red) and 90 K (purple) show the development of the I-O characteristics with temperature. Rate equation fits under the assumption of a purely 2D QW gain are shown for RT and 156 K data. The increased output intensity in the low excitation regime, due to contributions from localized states, is apparent at 156 K (shaded). Second-order autocorrelation measurements were performed at 156 K at the data points marked **i-iv**. A slope of 1 is indicated by the red dashed lines. The vertical offset was adjusted for clarity. **i-iv**, Autocorrelation traces taken at the excitation power densities marked in **a** display the characteristic bunching around threshold (**ii**, **iii**), which vanishes again for higher excitation (**iv**), indicating a transition towards Poissonian emission statistics in the lasing regime (shaded excitation range in **a**). The bunching in **i** cannot be fully resolved due to resolution limitation.

Instead of ideal spontaneous emission coupling, the thresholdless behavior can be ascribed to a transition in the gain material, with increased contributions in the low excitation range. This additional gain component stems from weakly localized states (0D) in the InGaN QW [30-31], that are activated below a localization temperature of ~ 160 K, as obtained from an Arrhenius evaluation of the QW emission (see also Figs. S3 and S4 and the related discussion in the SM). The result is a two component 0D-2D gain material. With falling temperature, thermal escape becomes less likely and the number of available localized states increases, leading to increased gain in the low excitation regime (until the

localized states are saturated) and eventually results in an inverse s-shaped I-O characteristic at low temperatures. In an intermediate temperature range around 156 K, these additional gain contributions exactly compensate the threshold nonlinearity. This interpretation is substantiated additionally by the observed discrepancies between the I-O data and the temperature dependent rate equation modeling under the assumption of a purely 2D gain (cf. Fig 3a), which can neither reproduce the I-O curve at 156 K, nor the inverse s-shaped intensity curve at low temperatures (cf. Fig. S1 in the SM and corresponding discussion). As including the second gain component into the rate equation modeling would involve the introduction of several unknown parameters, thereby severely limiting the significance of a fit, we refrain from an extended fitting including 0D and 2D gain materials. In the following we investigate the thresholdless I-O curve at 156 K (cf. Fig. 3). As for the room temperature case, the emission linewidth does not show any pronounced narrowing which could indicate the transition from spontaneous to stimulated emission. Obviously, a threshold can no longer be identified from the sole I-O characteristics. Thus, Fig. 3a is an excellent example demonstrating that a quantum-optical characterization is required to prove possible lasing in high- β nanolasers. We performed such measurements for four excitation power densities, as marked in Fig. 3a. Here, clear bunching at zero delay appears in the autocorrelation trace and vanishes as the excitation power is further increased (cf. Figs. 3i-iv). This unambiguously proves the transition in photon statistics at the onset of lasing and highlights the importance of a quantum-optical investigation to demonstrate lasing in the $\beta \rightarrow 1$ regime. In comparison to the room temperature measurements, we can only observe the far end of the bunching regime due to resolution limitation (cf. Fig. 3i). The timescale of the bunching in $g^{(2)}(\tau)$ is given by the characteristic timescale of intensity fluctuation of the emission [6-7], [25], which changes with temperature. The temperature dependent rate equation modeling suggests a decrease in the threshold excitation power density with respect to the room temperature measurements by a factor of ~ 4 (see SM section A).”

Through an Arrhenius type evaluation of the shift of the quantum well emission peak with temperature we are able to estimate a localization temperature of ~ 160 K (see also reworked section C of the SM). The temperature model for A and B coefficients suggests $A(160\text{K})/A(300\text{K})$ and $B(160\text{K})/B(300\text{K})$ are approximately 0.19 and 1.84, respectively (SI section A). The C coefficient remains approximately constant, when comparing RT and 160 K value (Table S3). A more quantitative analysis on the basis of a rate-equation analysis including both 0D and 2D gain components would involve the introduction of several new unknown parameters, severely limiting the significance of any obtained fit.

Reviewer:

With these comments in mind, I do not think that the authors have demonstrated the novelty or progress and clarity in presentation that is required for publication in Nature Communications.

Our response:

We hope that by our response and the revisions of the manuscript we convinced the reviewer about the novelty and clarity of presentation of our work and its suitability for publication in Nature Communications.

Reviewers' comments:

Reviewer #1 (Remarks to the Author):

I read the paper and the response from the authors carefully. I am happy to see the responses from the authors, and have a better understanding of the paper. Unfortunately, I am not convinced the authors are actually seeing a laser. From my own experience, second-order correlation of cavity enhanced PL from a quantum well becomes unity under higher power, while you observe a bunching behavior at lower power. The modeling of the laser is also performed with the rate equation, which is not valid for the thresholdless laser. A better quantum optical modeling with density matrix formalism should be used.

In recent past, there are too many papers on nano-lasers, which have used incorrect rate equation formalisms, and have attributed their observations to low-threshold lasing. The main problem is the signature of lasing at such low-power is model dependent, and all the known signatures of lasing are difficult to observe.

Reviewer #2 (Remarks to the Author):

The authors have responded clearly to my comments, thereby significantly improving the paper and making the context clear. The paper can in my opinion be published in its present form.

Reviewer #3 (Remarks to the Author):

This interesting manuscript reports investigations of the input-output characteristics of optically pumped GaN nanobeam "lasers" as a function of excitation level and lattice temperature. Hereby, power dependent photoluminescence measurements are compared with studies of the second-order photon correlation function $g_2(0)$ of the cavity emission. At elevated lattice temperatures, a gradual transition is observed in $g_2(0)$ from super-Poissonian ($g_2(0) > 1$) to near Poissonian ($g_2(0) \sim 1.1$) as the excitation level is increased. This gradual reduction of $g_2(0)$ towards a Poissonian level is indeed expected for nanolasers as the light-matter interaction makes a transition from the spontaneous to stimulated emission regime. The range of excitation power over which the transition occurs is accompanied by a weak non-linearity in the input-output characteristic and linewidth narrowing over a restricted range of pump power. While the results obtained are interesting and of a high technical quality, these two observations are insufficient to unambiguously demonstrate thresholdless lasing. Measurements performed at lower lattice temperatures (fig 3) exhibit much more complex input-output characteristics, including an "inverted s-shaped" curve – an effect identified as being due to the presence of gain regions with different dimensionalities and the impact of thermal redistribution of carriers. I have read the manuscript, referee reports and authors response with great interest.

Whilst the manuscript reports interesting results, I find that the data presented in the manuscript do not at all substantiate the claim that the results represent thresholdless lasing of nitride nanobeam cavities (at room temperature) and I cannot, therefore, recommend publication of the results in Nature Communications. Instead, the authors should consider publishing the data in a specialized journal targeted specifically at the community exploring nanolasers. In addition, I have some additional scientific comments on the manuscript.

- The linewidth broadening and redshift of the "lasing peak" observed in fig 2 at higher excitation levels are identified as arising from heating of the cavity region. This region where heating is observed overlaps precisely with the range over which $g_2(0)$ starts to reduce towards the Poissonian limit. Can the authors exclude that the observed reduction of the $g_2(0)$ does not simply

arise from a reduction of the coherence time due to heating that would manifest itself as weaker bunching in a CW HBT measurement with limited time resolution?

- The somewhat peculiar behavior of the temperature dependent input-output data presented in figure 3 is interpreted as reflecting carrier redistribution between the mixed dimensionality (0D and 2D) gain regions. Evidence for such mixed dimensionality gain regions should be present in e.g. temperature and power dependent luminescence / absorption spectra from the GaN basis material – can the authors show such data? The temperature dependent data shown in figure S3 do not unambiguously support this claim.
- The authors write at the beginning of the “Results” section “high- β nanolaser operate far from the thermodynamic limit, where $\beta^{-1} \rightarrow \infty$...” this would correspond to $\beta \rightarrow 0$... If the authors want to say this then I do not understand the sentence.

We thank the reviewers for their detailed evaluation of our manuscript and for their valuable comments and suggestions, which we have taken into account in order to improve the quality of the manuscript. In the following, we provide a point-to-point response to the questions/remarks of the reviewers. We indicate and summarize the changes and additions made to the manuscript.

Additionally, we have changed the title of the manuscript in order to better reflect the content of the paper. The new title is “*On thresholdless lasing features in high- β nitride nanobeam cavities: a quantum optical study*”.

Reviewer #1 (Remarks to the Author):

Reviewer:

I read the paper and the response from the authors carefully. I am happy to see the responses from the authors, and have a better understanding of the paper. Unfortunately, I am not convinced the authors are actually seeing a laser. From my own experience, second-order correlation of cavity enhanced PL from a quantum well becomes unity under higher power, while you observe a bunching behavior at lower power.

Our response:

We thank the reviewer for his/her positive feedback to our revisions and are glad to hear that he/she now has a better understanding of the paper. However, we are puzzled by his/her statement about the power dependence of the second-order correlation of cavity enhanced PL. In fact, to our opinion writing such an assertion (“*From my own experience [...]*”) without relevant references is of no value, since it can be used as a simple way to reject a paper based on a purely gratuitous assertion. In our opinion this is neither a fair, nor a scientifically correct treatment.

We would like to mention that a gain medium (“*quantum well*”) inside a resonator (“*cavity enhanced*”) is exactly the recipe for a laser, and as such it is of course possible that said change in emission statistics might be observed under high excitation – if the device makes the lasing transition.

To substantiate our case of lasing in the nanobeam cavity we studied another reference nanobeam with a lower quality Q factor, which does not show lasing. In the revised manuscript we clearly demonstrate that PL from such a cavity enhanced quantum well structure does not become unity under higher power - in contrast to the lasing structure and in contrast with the reviewer’s statement.

Action taken:

We added new data on a non-lasing nanobeam and discussed the difference between this structure and the lasing nanobeam to strengthen the proof of lasing and to rule out the concern of the reviewer.

Reviewer:

The modeling of the laser is also performed with the rate equation, which is not valid for the thresholdless laser. A better quantum optical modeling with density matrix formalism should be used.

Our response:

While we do not fully agree with the assertion made by the reviewer, in the sense that – if applied correctly – rate equations are able to at least properly describe the I-O characteristics in the high- β regime (e.g. discussed in [6, parts II & III]). Nonetheless, we took his/her criticism into account and saw the necessity to improve the theoretical description of our experimental results. To this end, we teamed up with experts on microscopic laser theories (F. Jahnke & C. Gies from the University of Bremen, Germany) and developed a semiconductor laser model for the given

structure. Importantly, the model gives us, in addition to the I-O characteristics, access to the photon statistics, the coherence time and carrier population functions – all of which substantiate our previous interpretation of lasing.

By applying this advanced model, we find excellent agreement with our experimental data, which now also include the characterization of a non-lasing nanobeam as reference. We achieve a coherent description of the RT data (lasing structure and non-lasing structure), as well as the low temperature data, with a small common set of parameters.

Action taken:

We developed the microscopic model and revised the manuscript in the following way (excerpts):

Page 2:

Our measurements are complemented with a microscopic laser theory to access simultaneously the I-O characteristic, zero time delay photon autocorrelation function $g^{(2)}(0)$, coherence time of the emission, and the carrier population functions. This combination provides access to the underlying lasing physics, in particular to the “ideal” autocorrelation function that is not detector limited. By combining the calculated coherence times with the detector resolution we can simulate the measured autocorrelation function in excellent quantitative agreement with our experimental data.

Pages 4-5:

The following evaluation is accompanied by the results of a microscopic laser theory for the interaction between the two-dimensional QW gain material and the fundamental cavity mode. Coupled equations for quantum-mechanical expectation values are solved, describing the wave vector dependent electron and hole populations, the intracavity photon number, as well as correlation functions that connect carrier and photonic degrees of freedom. Including these correlation functions provides direct access to the zero time delay photon autocorrelation function $g^{(2)}(0)$, a quantity that is inaccessible in a rate equation-based analysis. Calculating the first-order coherence function $g^{(1)}(\tau)$ - and with it the coherence time - allows us to model the decay of $g^{(2)}(\tau)$ from its zero time delay value in order to simulate the detector-limited time resolution of the experiment. We would like to emphasize that excellent simultaneous agreement with all experimental data (I-O and $g^{(2)}(0)$ of laser and reference structure) is achieved based on a single set of parameters that enter the model (cf. Table S1 in the supplementary material (SM)). Further details of the optical characterization are provided in the methods summary and the theoretical model is described in detail in the SM.

Fig. 2 and accompanying discussion (excerpt herein):

In Fig. 2 we compare room temperature characteristics representative of a lasing and a non-lasing nanobeam cavity, which serves as a reference. The main difference between both structures lies in the Q-factor, which has been determined to $Q \sim 2200$ for the nanobeam laser and $Q \sim 1800$ for the reference nanobeam. The room temperature I-O curve of the nanobeam laser is depicted in Fig. 2a, together with the cavity mode below threshold (inset). The solid line is obtained from the microscopic model, assuming a non-radiative loss rate $A_{nr} = 5 \times 10^7 \text{ s}^{-1}$, and exhibits a slight threshold nonlinearity before converging to a slope of 1 (dashed line). Due to the strong guiding of photons into the lasing mode, inherent to the nanobeam geometry, emission into non-lasing modes is largely suppressed. This is reflected in a β -factor of ~ 0.7 for both nanobeams, which can be estimated within the scope of the microscopic model along the lines of [10], taking into account the light-matter coupling strength, as well as radiative losses (cf. Table S1). When compared to a rate equation analysis, the ratio of spontaneous emission into the lasing mode is calculated directly, meaning that the β -factor is no longer an input-parameter (fit parameter) to the theory (see corresponding SM section). Nonetheless, the obtained value is in good agreement with the results of a rate equation analysis of the nanobeam laser ($\beta_{RE} = 0.7 \pm 0.2$) using model and parameters employed in [13].

[...]

Figure 2 | Room temperature optical and quantum-optical characterization of a lasing (left) and a non-lasing (right) III-nitride nanobeam cavity. *a, d*, Room temperature I-O curves. The theoretical model (solid line) in *a* shows a slight nonlinearity before converging to a slope of 1 (indicated by the dashed line). Inset in *a*: Fundamental cavity mode at 0.64 kW/cm². The I-O characteristic in *d* is governed by non-radiative losses and does not show an s-bend before saturating. The increased output intensity with respect to *a* is indicative of an increase in light scattering towards the vertical direction. Note that the intracavity photon number is higher for the lasing nanobeam, which allows building up a coherent photon population. *b, e*, Resonance peak wavelength (green) and linewidth (FWHM, black). Above about 10 kW/cm² the development of resonance wavelength and emission linewidth is dominated by heating of the cavity region. The lasing structure in *b* exhibits a slight decrease in linewidth around $P \approx 5$ kW/cm². *c, f*, Second-order autocorrelation function at zero time delay as obtained from experiment (data points) and theory. Proof of the transition to coherent emission (shaded excitation range) is provided by the power dependence of the deconvolved second-order autocorrelation data, showing a clear trend towards the Poisson limit ($g^{(2)}(0) = 1$) with increasing excitation power density. In contrast, the power dependence in *f* reveals a constant $g^{(2)}(0) \leq 1.2$. The evolution of the photon statistics is well reproduced by the microscopic theory (ideal: solid line, convoluted: dashed line), when taking into account the calculated coherence time and the convolution with the temporal resolution (~ 225 ps) of the HBT setup.

Page 7:

For the nanobeam laser, theory predicts a clear transition from thermal emission to lasing (solid line in Fig. 2c). The transition is accompanied by an increase in coherence time from ~ 1 ps to ~ 800 ps across the transition region (cf. Fig. S1 in the SM). Since the decay of the autocorrelation function $g^{(2)}(\tau)$ with respect to τ is related to the coherence time, the zero time delay value can only be resolved if the coherence time exceeds the detector resolution. This

becomes apparent in the convoluted $g^{(2)}(0)$ trace (dashed line in Fig. 2c), which excellently reproduces the experimental results over the excitation range. For the reference nanobeam (Fig.2 f), the experimentally observed constant value of ~ 1.2 is traced back to an only moderately increasing coherence time, combined with a calculated $g^{(2)}(0)$ that stays largely thermal until saturation is reached.

Reviewer:

In recent past, there are too many papers on nano-lasers, which have used incorrect rate equation formalisms, and have attributed their observations to low-threshold lasing. The main problem is the signature of lasing at such low-power is model dependent, and all the known signatures of lasing are difficult to observe.

Our response:

In our opinion, the criticism raised here – that many recent papers did not conclusively prove lasing in nanolasers – was already discussed at length in our first response. It can hardly be seen as our fault that these papers ‘sloppily’ dealt with the characterization of high- β lasing. We agree that a rate equation analysis critically depends on the chosen model and (pending a proper documentation) cannot be central to the investigation of high- β lasing. It should merely be used in supplement to an optical and quantum-optical characterization. The shortcomings of an analysis based on a (realistic) rate equation model (including background absorption) are discussed in ref. [6, parts II & III] and compared to a quantum statistical theory (birth-death model). The main drawback of a rate equation analysis is their inability to properly reproduce the statistical aspects of the emission. In contrast, a quantum statistical or microscopic model can be used as a more potent and accurate tool. In order to substantiate our results, we developed and added such a theory to the paper and obtain excellent agreement with our measurements (see our answer above). The negligence of many recent papers is discussed in detail in the manuscript and was part of our motivation to conduct a thorough analysis including a quantum-optical characterization.

We also agree that the identification of high- β lasing is more difficult when compared to “classical” lasers, like e.g. VCSELs, as the conventionally sought-after signatures (I-O non-linearity and substantial linewidth decrease) are far less pronounced. However, this is exactly why a thorough investigation of the underlying photon statistics is crucial, as has been discussed previously, e.g. [6-7], [9], [23-24] and the reason why we measured the photon statistics alongside classical indicators. The transition in photon statistics is (if it can be resolved) widely accepted as a hard criterion for the lasing transition (from thermal to coherent emission, i.e. spontaneous to dominantly stimulated emission) in the nanolasers and quantum optics community.

With the addition of the microscopic model, we feel that our work now adds a significant value to the general field of novel laser devices and low-threshold high- β lasers – in particular in light of the reviewer’s statement - by showing how to carefully isolate and identify signatures *beyond* a rate equation analysis.

Action taken:

We revised the manuscript to highlight and discuss these important aspects.

Page: 3:

The effort to develop low power consuming, i.e. low threshold, nanoscale lasers usually goes hand in hand with the quest to achieve high- β lasing. Interestingly though, high- β nanolasers do not exhibit an abrupt, phase transition-like, lasing threshold [6]. Instead, high- β lasing entails a gradual change in emission properties, including output intensity, linewidth and the transition from thermal to coherent emission, over a wide range of excitation powers [6-9], [23-26]. High- β nanolasers should thus not be approached as conventional lasers, but additionally through statistical properties of the emitted radiation [6-7]. We would like to emphasize that the concept of “thresholdless lasing”, associated with $\beta = 1$ and the absence of non-radiative losses [8], does not imply a threshold at zero excitation, a concept developed in an early publication [4]. Instead, the gradual transition towards coherent emission always occurs at finite excitation and is thus visible in excitation dependent second-order autocorrelation

measurements [6-7], [9]. In practice, most publications on high- β lasers still rely solely on I-O characteristics in combination with rate equation fitting [27], in particular when it comes to the study of nanolasers operating at elevated temperatures [13], [20-22], [28]. Thereby, coherence and statistical properties of the emission, which cannot be captured using rate equation modeling, are neglected [6]. In order to preserve a reliable and practically meaningful definition for a lasing threshold, it was repeatedly proposed to rely on statistical properties of the emitted radiation [6-7], [9], [23-24].

Pages 5-6:

It is important to note that soft nonlinearities in the I-O curve, similar to that observed in Fig. 2a, can in principle also be related to trap filling [30] and thus cannot give an unambiguous proof of lasing. In fact, the observation of further indications of stimulated emission is required to prove lasing, in particular towards the limit of a “thresholdless” laser with a linear I-O curve. Another conventional signature of the onset of lasing is a decrease in the emission linewidth at half maximum (FWHM) and an associated increase in temporal coherence at the transition from predominantly spontaneous emission to stimulated emission. In this context, a power dependent linewidth narrowing can also be caused by quenching of absorption losses, which complicates the correct interpretation of this lasing indicator in high- β lasers. In contrast to conventional lasers like e.g. vertical cavity surface emitting lasers, where $\beta \ll 1$, high- β nanolasers typically show only minor changes in linewidth at threshold [2-3], [23], [27]. This is a result of increased refractive index fluctuations (gain-refractive index coupling) in the excitation range around threshold [26]. Through the soft onset of lasing, which takes place already at low carrier densities and with few intracavity photons [6], these fluctuations persist over a large range of excitation powers, leading to an almost constant linewidth in the threshold region [26]. Considering the discussion above, a lasing threshold could be falsely identified from these classical indicators.

Page 6:

As the conventionally sought-after lasing signatures (a pronounced I-O nonlinearity and linewidth decrease) are far more elusive, unambiguous proof of the onset of stimulated emission in high- β nanolasers requires excitation power dependent second-order autocorrelation measurements in order to monitor the change in emission statistics.

Reviewer #2 (Remarks to the Author):

Reviewer:

The authors have responded clearly to my comments, thereby significantly improving the paper and making the context clear. The paper can in my opinion be published in its present form.

Our response:

We are glad to read that we could improve the manuscript and clarify its message. We would like to thank the reviewer again for his/her comments/questions regarding the original version of the manuscript, which served as a valuable guideline for our improvements.

Reviewer #3 (Remarks to the Author):

Reviewer:

This interesting manuscript reports investigations of the input-output characteristics of optically pumped GaN nanobeam “lasers” as a function of excitation level and lattice temperature. Hereby, power dependent photoluminescence measurements are compared with studies of the second-order photon correlation function $g_2(0)$ of the cavity emission. At elevated lattice temperatures, a gradual transition is observed in $g_2(0)$ from super-Poissonian ($g_2(0) > 1$) to near Poissonian ($g_2(0) \sim 1.1$) as the excitation level is increased. This gradual reduction of $g_2(0)$ towards a Poissonian level is indeed expected for nanolasers as the light-matter interaction makes a transition from the spontaneous to stimulated emission regime. The range of excitation power over which the transition occurs is accompanied by a weak non-linearity in the input-output characteristic and linewidth narrowing over a restricted range of pump power. While the results obtained are interesting and of a high technical quality, these two observations are insufficient to unambiguously demonstrate thresholdless lasing. Measurements performed at lower lattice temperatures (fig 3) exhibit much more complex input-output characteristics, including an “inverted s-shaped” curve – an effect identified as being due to the presence of gain regions with different dimensionalities and the impact of thermal redistribution of carriers. I have read the manuscript, referee reports and authors response with great interest.

Whilst the manuscript reports interesting results, I find that the data presented in the manuscript do not at all substantiate the claim that the results represent thresholdless lasing of nitride nanobeam cavities (at room temperature) and I cannot, therefore, recommend publication of the results in Nature Communications. Instead, the authors should consider publishing the data in a specialized journal targeted specifically at the community exploring nanolasers.

Our response:

We thank the reviewer for expressing his/her interest in our results and are confident to be able to resolve open questions and the concerns raised. We hope that in doing so we can convince the reviewer of the general interest of our results to the broad readership of Nature Communications.

We agree with the reviewer, in saying that:

This gradual reduction of $g_2(0)$ towards a Poissonian level is indeed expected for nanolasers as the light-matter interaction makes a transition from the spontaneous to stimulated emission regime.

Investigating the transition in photon statistics has indeed become an established method to identify the lasing transition in high- β nanolasers, e.g. [6-7, 9, 23-25]. However, the reviewer might have misunderstood the discussion of our room temperature results when saying:

[...] these two observations are insufficient to unambiguously demonstrate thresholdless lasing.

and

[...] I find that the data presented in the manuscript do not at all substantiate the claim that the results represent thresholdless lasing of nitride nanobeam cavities (at room temperature) [...]

We do not claim to observe thresholdless lasing at room temperature, but high- β lasing with $\beta \sim 0.7$. The term thresholdless refers to the I-O curve at 156 K, where we discuss it in terms of a two-component gain medium. In light of the achieved β -value of 0.7 the linear I-O characteristic is highly unexpected. This is a very important aspect and message for the nanolasers community, that one should be careful when associating a linear I-O curve with $\beta = 1$ – even when complemented with a quantum-optical characterization – as it could be the result of a complex gain redistribution as we demonstrate by temperature dependent studies. In doing so, we show that a temperature dependent investigation is necessary to fully explore the device characteristics. Please note that the focus of the

revised manuscript was shifted away from the discussion of the thresholdless I-O characteristic towards a more detailed analysis of the room temperature characterization. To this end, we added the characterization of a non-lasing nanobeam as reference. Additionally, we replaced the accompanying rate equation analysis with an involved microscopic model that allows us to also access the photon statistics. An in-depth response, supplemented by changes to the manuscript can be found in direct response to the bullet points below.

Reviewer:

In addition, I have some additional scientific comments on the manuscript.

- *The linewidth broadening and redshift of the “lasing peak” observed in fig 2 at higher excitation levels are identified as arising from heating of the cavity region. This region where heating is observed overlaps precisely with the range over which $g_2(0)$ starts to reduce towards the Poissonian limit. Can the authors exclude that the observed reduction of the $g_2(0)$ does not simply arise from a reduction of the coherence time due to heating that would manifest itself as weaker bunching in a CW HBT measurement with limited time resolution?*

Our response:

The reviewer is correct in saying that a reduction in coherence time due to heating at higher excitation could mimic an excitation dependent reduction of the zero time delay bunching. In order to investigate a potential excitation dependent thermal influence on the $g^{(2)}(\tau)$ measurements, additional measurements on a non-lasing nanobeam were performed (see revised Fig. 2). Wavelength and linewidth (Fig. 2e) exhibit a similar, thermally dominated behavior with increasing excitation power density when compared to the nanobeam laser (Fig. 2b). The extracted zero-delay autocorrelation value (Fig. 2f), however, remains at a constant value of $g^{(2)}(0) \sim 1.2$ and does not show a thermal influence. We believe this observation reasonably substantiates the claim, that the decrease in $g^{(2)}$ originates from a transition in photon statistics from thermal towards coherent emission. Note also, that the superimposed decrease in linewidth (Fig. 2b) is not observed for the non-lasing structure. In addition, the microscopic model, that has been developed in cooperation with experts from the University of Bremen (F. Jahnke and C. Gies), does predict the transition in photon statistics in the same range of excitation power densities. Based on a small common set of parameters, the model is found to be in excellent agreement with the obtained data. The whole result section has been reworked to incorporate and discuss the new measurements and theory.

We believe that the combined investigation of nanobeam laser and reference cavity, in excellent agreement with the microscopic model, greatly substantiates our previous observations and interpretation.

Action taken:

We greatly revised the results section of the manuscript.

Fig. 2 and accompanying discussion:

In Fig. 2 we compare room temperature characteristics representative of a lasing and a non-lasing nanobeam cavity, which serves as a reference. The main difference between both structures lies in the Q-factor, which has been determined to $Q \sim 2200$ for the nanobeam laser and $Q \sim 1800$ for the reference nanobeam. The room temperature I-O curve of the nanobeam laser is depicted in Fig. 2a, together with the cavity mode below threshold (inset). The solid line is obtained from the microscopic model, assuming a non-radiative loss rate $A_{nr} = 5 \times 10^7 \text{ s}^{-1}$, and exhibits a slight threshold nonlinearity before converging to a slope of 1 (dashed line). Due to the strong guiding of photons into the lasing mode, inherent to the nanobeam geometry, emission into non-lasing modes is largely suppressed. This is reflected in a β -factor of ~ 0.7 for both nanobeams, which can be estimated within the scope of the microscopic model along the lines of [10], taking into account the light-matter coupling strength, as well as radiative losses (cf. Table S1). When compared to a rate equation analysis, the ratio of spontaneous emission into the lasing mode is calculated

directly, meaning that the β -factor is no longer an input-parameter (fit parameter) to the theory (see corresponding SM section). Nonetheless, the obtained value is in good agreement with the results of a rate equation analysis of the nanobeam laser ($\beta_{RE} = 0.7 \pm 0.2$) using model and parameters employed in [13].

[...]

In the present case, we observe a minor decrease in the emission linewidth, superimposed on a dominantly thermal behavior, at $P \approx 5 \text{ kW/cm}^2$, cf. Fig. 2b. Linewidth and emission wavelength of the reference cavity (cf. Fig. 2e) show an overall similar, thermally dominated, trend. A decreasing linewidth at intermediate excitation power densities is not observed.

[...]

Figure 2 | Room temperature optical and quantum-optical characterization of a lasing (left) and a non-lasing (right) III-nitride nanobeam cavity. a, d, Room temperature I-O curves. The theoretical model (solid line) in a shows a slight nonlinearity before converging to a slope of 1 (indicated by the dashed line). Inset in a: Fundamental cavity mode at 0.64 kW/cm^2 . The I-O characteristic in d is governed by non-radiative losses and does not show an s-bend before saturating. The increased output intensity with respect to a is indicative of an increase in light scattering towards the vertical direction. Note that the intracavity photon number is higher for the lasing nanobeam, which allows building up a coherent photon population. b, e, Resonance peak wavelength (green) and linewidth (FWHM, black). Above about 10 kW/cm^2 the development of resonance wavelength and emission linewidth is dominated by heating of the cavity region. The lasing structure in b exhibits a slight decrease in linewidth around $P \approx 5 \text{ kW/cm}^2$. c, f, Second-order autocorrelation function at zero time delay as obtained from experiment (data points) and theory. Proof of the transition to coherent emission (shaded excitation range) is provided by the power dependence of the deconvolved second-order autocorrelation data, showing a clear trend towards the Poisson

limit ($g^{(2)}(0) = 1$) with increasing excitation power density. In contrast, the power dependence in f reveals a constant $g^{(2)}(0) \leq 1.2$. The evolution of the photon statistics is well reproduced by the microscopic theory (ideal: solid line, convoluted: dashed line), when taking into account the calculated coherence time and the convolution with the temporal resolution (~ 225 ps) of the HBT setup.

The results of an excitation power dependent investigation of the photon statistics are shown in Fig. 2c, f. In order to obtain the zero time delay value $g^{(2)}(0)$, the measured autocorrelation traces were fitted using a convolution of the idealized fitting function and the detector response, taking into account the temporal resolution $\Delta t_{res} \approx 225$ ps of the HBT setup. See also Fig. S3 and corresponding explanations in the SM for further details on the fitting procedure. For the lasing nanobeam, we observe clear bunching behavior ($g^{(2)}(0) > 1$), which becomes less pronounced with increasing excitation power density, indicating the transition from spontaneous to dominantly stimulated emission of light (Fig. 2c). The deduction of a high β value is supported by a smeared out and incomplete transition to the Poisson limit within the investigated excitation power density range [25]. The deconvolved data show a bunching maximum of $g^{(2)}(0) \approx 1.4$ in the threshold region. In contrast, in case of the non-lasing cavity, $g^{(2)}(0)$ remains constant ($g^{(2)}(0) \sim 1.2$) over the investigated range of excitation power densities (Fig. 2f). A thermal influence on the photon statistics via a reduced coherence time, is not observed. For the nanobeam laser, theory predicts a clear transition from thermal emission to lasing (solid line in Fig. 2c). The transition is accompanied by an increase in coherence time from ~ 1 ps to ~ 800 ps across the transition region (cf. Fig. S1). Since the decay of the autocorrelation function $g^{(2)}(\tau)$ with respect to τ is related to the coherence time, the zero time delay value can only be resolved if the coherence time exceeds the detector resolution. This becomes apparent in the convoluted $g^{(2)}(0)$ trace (dashed line in Fig. 2c), which excellently reproduces the experimental results over the excitation range. For the reference nanobeam (Fig. 2f), the experimentally observed constant value of ~ 1.2 is traced back to an only moderately increasing coherence time, combined with a calculated $g^{(2)}(0)$ that stays largely thermal until saturation is reached. Furthermore, we find spectral hole-burning at the cavity-mode energy in the calculated non-equilibrium carrier distribution functions shown in the SM (cf. Fig. S2a), which is another indicator for lasing operation [10].

Finally, we point out that a mean intracavity photon number of one is not necessarily a signature of lasing [6, 31]. The theoretical results for both the laser and the reference nanobeam exhibit photon numbers above one, but by comparing Fig. 2a and c, one can infer that a coherent photon population builds up only above a mean photon number of 100 [31]. It is the imbalance of spontaneous towards stimulated emission that makes the emission coherent above threshold, and this imbalance is not only determined by the mean intracavity photon number, but also by the inversion of the system that is expressed by the non-equilibrium distribution functions for electrons and holes.

Reviewer:

- The somewhat peculiar behavior of the temperature dependent input-output data presented in figure 3 is interpreted as reflecting carrier redistribution between the mixed dimensionality (0D and 2D) gain regions. Evidence for such mixed dimensionality gain regions should be present in e.g. temperature and power dependent luminescence / absorption spectra from the GaN basis material – can the authors show such data? The temperature dependent data shown in figure S3 do not unambiguously support this claim.

Our response:

The temperature dependence of the photoluminescence emission energy of the InGaN quantum well (QW) shown in Fig. S4 (former Fig. S3) exhibits an s-shape, i.e., an initial redshift followed by a blueshift and finally a redshift coinciding with the decrease in energy of the bandgap of the transition energy. Such a behavior is the recognized

signature of the transition from a regime where carriers are initially localized (quantum dot or OD-like regime) to a regime where carriers are fully delocalized thanks to the thermal energy which allows overcoming potential fluctuations (2D-like regime). It has been widely documented in numerous articles over the past few decades and it is a commonly observed signature in indium-containing alloys belonging to the III-nitride family (see, e.g., Ref. 9 of the Supplementary Material, Y.-H. Cho et al., APL **73**, 1370-1372 (1998), the first historical report on this behavior in III-N QWs, a highly cited work as can be checked on Web of Science (> 470 citations)), which also partly explains the large insensitivity of the InGaN alloy to nonradiative recombination and hence its suitability for the realization of high internal quantum efficiency optoelectronic devices such as blue and white light-emitting diodes that led to the Nobel Prize in Physics in 2014. Therefore, we do not believe that it is necessary to describe once more the physics behind such a dependence. In addition, the requested absorption spectra are extremely difficult to measure on InGaN quantum wells, if not impossible, due to the large inhomogeneous broadening typical of such samples as explained, e.g., in M. Glauser et al., J. Appl. Phys. **115**, 233511 (2014). This is even truer for the present sample owing to the nature of the substrate, silicon, which is absorbing light at the QW emission energy, hence preventing any transmission-like measurements.

Reviewer:

- *The authors write at the beginning of the “Results” section “high- β nanolaser operate far from the thermodynamic limit, where $\beta^{-1} \rightarrow \infty \dots$ ” this would correspond to $\beta \rightarrow 0 \dots$. If the authors want to say this then I do not understand the sentence.*

Our response:

With this sentence we tried to convey that high- β nanolasers operate outside of the thermodynamic limit (they have $\beta \rightarrow 1$) and that the thermodynamic limit in terms of the β -factor means $\beta \rightarrow 0$, which usually holds for “classical” lasers like e.g. VCSELs (typically $\beta < 10^{-5}$), c.f. Ref. [6]. We understand that the statement has been misleading and we revised the text to clarify this point.

Action taken:

The sentence has been rewritten, it now reads:

Interestingly though, high- β nanolasers do not exhibit an abrupt, phase transition-like, lasing threshold [6].

Reviewers' comments:

Reviewer #1 (Remarks to the Author):

I am not convinced that the authors are actually observing a lasing behavior. The laser checklist they filled up is very confusing and possibly misleading. For example, when the authors responded to how many devices have been measured, they say 1000. Does this mean 1000 devices are lasing? I highly doubt that, but if that is the case, that is actually a very strong result. If not, then stating this is clearly misleading. Also, when the authors say that they have measured spatial mode profile, why do not they report it in the paper/ supplement, without saying that it is in a 2015 nano letter paper? Overall, I think the laser checklist is very misleading, and in no way, it shows that there is actual sign of lasing.

Also in figure 2, the g_2 for the lasing device and non-lasing device at the highest power are similar. I agree that there is a different trend, but why do not the authors increase the power to observe actually $g_2=1$? Hence, in my opinion, the paper does not provide enough advancement to warrant publication in Nature Communication.

Reviewer #4 (Remarks to the Author):

The manuscript, titled "On thresholdless lasing features in high- β nitride nanobeam cavities: a quantum optical study", by Stefan T. Jagsch et al. reports a high- β GaN-based photonic crystal nanobeam cavity laser with the experimental data and theoretical analysis. The study investigated the input-output characteristics of the GaN-based photonic crystal cavity under optically pumping conditions. The results are interesting. However, based on the content presented in the current manuscript, I am not 100% convinced that the current version of the manuscript can be published in Nature Communication. The comments are followed

Pros:

- 1) The excellent theoretical modeling and physical interpretation for the photonic nanocavity lasers.
- 2) The nice InGaN/GaN photonic crystal nanobeam cavity demonstrated on the silicon substrate.

Cons:

- 1) While the authors report some interesting analysis and issues on the photonic crystal nanobeam cavity, the main concern for the manuscript is the experimental data of the GaN nanobeam cavity shown in the current manuscript are not enough to claim the lasing action from the photonic crystal cavity.

-- In the semiconductor text books or some literatures related to photonic crystal lasers, the linewidth narrowing behavior around the threshold is one of key signature to verify the lasing action. It is also well known that the laser linewidth is described by the Schawlow-Townes formula. Actually the laser linewidth is normally $(1 + \alpha^2)$ times wider than this formula with semiconductor gain materials, where α is the linewidth enhancement factor. The linewidth data describe in the manuscript did not show or follow the behaviors of the small semiconductor lasers. Authors might also illustrate if the conventional physical descriptions are not enough for the nanocavity.

-- To verify the coupling between the GaN emission and the resonant mode of the photonic crystal nanobeam cavity, authors should show the PL spectrum aligned with the whole lasing spectrum.

- 2) The LL turn-on curve from the photonic crystal cavity did not show the obvious roll-over behavior due to the thermal effects under the high power pumping. Why authors attribute the

unusual linewidth increase of the spectrum linewidth to the heating of the cavity under higher power pumping?

3) I also notice that there is only one InGaN/GaN QW in the photonic crystal laser cavity which is quite different to most of photonic crystal lasers. However, to support enough gain within such small cavity and overcome the optical loss within such compact cavities, 3-5 pairs of quantum wells embedded in the photonic crystal cavity is the most common vertical structure. What are the advantages for this 1 pair QW in the GaN nanobeam laser cavity?

We thank the reviewers for their evaluation of our manuscript and their valuable comments and suggestions, which we have taken into account in order to further improve both quality and clarity of the manuscript. In the following, we provide a point-to-point response to the questions/remarks of the reviewers. We indicate and summarize the changes and additions made to the manuscript.

Reviewer #1 (Remarks to the Author):

Reviewer:

I am not convinced that the authors are actually observing a lasing behavior. The laser checklist they filled up is very confusing and possibly misleading. For example, when the authors responded to how many devices have been measured, they say 1000. Does this mean 1000 devices are lasing? I highly doubt that, but if that is the case, that is actually a very strong result. If not, then stating this is clearly misleading. Also, when the authors say that they have measured spatial mode profile, why do not they report it in the paper/ supplement, without saying that it is in a 2015 nano letter paper? Overall, I think the laser checklist is very misleading, and in no way, it shows that there is actual sign of lasing.

Our response:

We thank the reviewer for his/her additional comments, this time concerning the *laser checklist*. It is surprising to us that the reviewer considers the filled checklist as “*very misleading, and in no way, it shows that there is actual sign of lasing*”. As stressed already in response to the previous reviews, the identification of lasing in a high- β nanolaser is rather intricate and often relies on a combination of different criteria [see C. Gies *et al.* in “*Handbook of optoelectronic device modeling and simulation*”, J. Piprek Ed., CRC Press, Taylor & Francis Group (2017)]. Measurements of the autocorrelation function have, therefore, become an established criterion in the literature to identify the transition to coherent emission by its evolution towards 1 from higher values. This behavior is observed for the nanobeam laser but not for the reference nanobeam. In addition to this strong experimental proof, we have complemented our work by a laser theory that provides input-output characteristics, autocorrelation function, coherence time, and carrier population functions. For one consistent parameter set (only differing cavity-Q for the lasing and non-lasing nanobeam), the observed behavior is reproduced by the model. Importantly, theory reproduces the measured $g^{(2)}$ by taking into account the calculated coherence time in the convolution of the calculated $g^{(2)}$ -function. The *simultaneous* agreement of these three quantities that, as we point out again, are not obtained independently, but from one consistent calculation, together with the measured $g^{(2)}$ approaching one, gives a very strong account of lasing operation.

Concerning the second point, we have clarified how many devices have been investigated in a new section in the manuscript, see below. Furthermore, a picture of the mode emission was added to the SM. We would like to note in this context, that the development of a speckle pattern, or more generally interference effects, in the spectrum is a field coherence (first-order coherence) effect and is strictly speaking not a proof for lasing. Field coherence can indeed be introduced by simply passing light through a narrow bandpass (as used in e.g. Hayenga *et al.* *Optica* **3**, 1187-1193 (2016)) or a spatial filter. To this end, a nanocavity (stopband and resonance) can constitute a narrow bandpass filter.

Action taken:

A section on the number of measured devices has been added to the methods section of the main manuscript.

Page 11:

More than 1000 nanobeam cavities have been fabricated as part of a processing optimization series (not all cavities are nominally identical), investigating the impact of sample processing on the target parameters based on 3D-FDTD

calculations. Approximately 200 nanobeams have then been pre-characterized regarding cavity resonance wavelength, quality factor and overall output intensity. Subsequently, I-O characteristics have been recorded for 10 nanobeams in the sample region that proved to be the most promising after pre-characterization. Of these nanobeam cavities, 5 showed indications of a lasing transition in the I-O characteristics. Room temperature power dependent second-order autocorrelation measurements have been carried out for 2 lasing and 1 non-lasing (reference) nanobeam.

Pictures of the mode emission below and above threshold were added to the SM.

SM pages 14-15:

E. Optical images of the emission

Figure S8 shows optical images of the nanobeam emission that have been collected for different excitation power densities. Excitation laser and GaN emission have been filtered out using an appropriate low-pass filter. Background emission from the QW gain material is collected as well. With increasing excitation power density we observe enhanced directionality of the emission perpendicular to the nanobeam, in agreement with [16]. The appearance of a 'stripe pattern' (cf. detail in Fig. S8) is attributed to the structure itself. We would like to note in this context that the appearance of speckles, fringes or generally interference effects in optical images of a device's emission cannot be seen as (conclusive) evidence for lasing. They are a field coherence effect (first-order coherence) and can also be produced by passing light through a narrow bandpass [17] or spatial filter (e.g. double slit experiment by Thomas Young).

Figure S8 | Optical images of the nanobeam emission. Room temperature I-O curve (panel a of Fig. 2 in the main text) along with optical images of the emission. We observe an increased directionality of the emission with increasing excitation. Detail: We attribute the appearance of a slight 'stripe pattern' parallel to the nanobeam to the structure itself.

Reviewer:

Also in figure 2, the g^2 for the lasing device and non-lasing device at the highest power are similar. I agree that there is a different trend, but why do not the authors increase the power to observe actually $g^2=1$? Hence, in my opinion, the paper does not provide enough advancement to warrant publication in Nature Communication.

Our response:

Many previous publications have put thought into the interpretation of $g^{(2)}(0)$. When measured with finite time resolution, the trend pointed out by the reviewer is not a marginality but a very important feature. For the lasing nanobeam, the **initial increase followed by a decrease** actually gives a stronger account of the transition to lasing than the hard value of 1. It demonstrates that an increase of coherence time takes place that allows the measurement setup, at least partly, to resolve the bunching before the threshold. Secondly, it means that the following decrease really comes from the decrease of the actual $g^{(2)}(0)$ function as it approaches the Poissonian limit of 1.

In experiment, we were limited to a maximum achievable excitation power density of about 25 kW cm^{-2} of the available laser system. According to theory we expect $g^{(2)}(0) = 1$ at about 100 kW cm^{-2} (the calculated $g^{(2)}(0)$ -value is 1.009 at 100 kW/cm^2), which is not accessible by the laser in the given experimental configuration. We would like to note that independent of the available laser power strong heating and a possibly resulting degradation of the sample could prevent us from exploring the photon statistics in the very high excitation power range, where we expect $g^{(2)}(0) = 1$. At the same time we would like to point out that smallest deviations in the actual photon statistics p_n from a Poissonian distribution can cause $g^{(2)}(0)$ being either 1 or 1.1. While a value of 1 can be expected in the lasing regime of macroscopic devices, nanolasers that do not operate with a lot of excess gain to cross the threshold may be subject to fluctuations above threshold that can prevent $g^{(2)}(0)$ from reaching the ideal value of 1. In this case, it is not a matter of stronger pumping, but a consequence of attaining lasing in a nanoscale device where stimulated emission may be insufficient to completely overwhelm spontaneous emission.

Action taken:

The section discussing the room temperature second-order autocorrelation measurements has been amended. The new paragraph reads:

page 7:

For the lasing nanobeam, we observe clear bunching behavior ($g^{(2)}(0) > 1$), which becomes less pronounced with increasing excitation power density, indicating the transition from spontaneous to dominantly stimulated emission of light (Fig. 2c). The deduction of a high β value is supported by a smeared out and incomplete transition to the Poisson limit within the available excitation power density range [25]. While the $g^{(2)}(0)$ -signature of a fully Poissonian photon statistic is not observed in experiment, theory suggests that $g^{(2)}(0)$ approaches 1 at $\sim 100 \text{ kW cm}^{-2}$. Operation at such excitation power densities would require improved thermal properties of the nanobeam cavities in order to reduce sample heating. The deconvolved data show a bunching maximum of $g^{(2)}(0) \approx 1.4$ in the threshold region and $g^{(2)}(0) \approx 1.1$ at high excitation. In contrast, in case of the non-lasing cavity, $g^{(2)}(0)$ remains constant at a value of ~ 1.2 over the investigated range of excitation power densities (Fig. 2f).

Reviewer #4 (Remarks to the Author):

Reviewer:

The manuscript, titled "On thresholdless lasing features in high- β nitride nanobeam cavities: a quantum optical study", by Stefan T. Jagsch et al. reports a high- β GaN-based photonic crystal nanobeam cavity laser with the experimental data and theoretical analysis. The study investigated the input-output characteristics of the GaN-based photonic crystal cavity under optically pumping conditions. The results are interesting. However, based on the content presented in the current manuscript, I am not 100% convinced that the current version of the manuscript can be published in Nature Communication. The comments are followed

Pros:

- 1) The excellent theoretical modeling and physical interpretation for the photonic nanocavity lasers.*
- 2) The nice InGaN/GaN photonic crystal nanobeam cavity demonstrated on the silicon substrate.*

Cons:

- 1) While the authors report some interesting analysis and issues on the photonic crystal nanobeam cavity, the main concern for the manuscript is the experimental data of the GaN nanobeam cavity shown in the current manuscript are not enough to claim the lasing action from the photonic crystal cavity.*

-- In the semiconductor text books or some literatures related to photonic crystal lasers, the linewidth narrowing behavior around the threshold is one of key signature to verify the lasing action. It is also well known that the laser linewidth is described by the Schawlow-Townes formula. Actually the laser linewidth is normally $(1 + \alpha^2)$ times wider than this formula with semiconductor gain materials, where α is the linewidth enhancement factor. The linewidth data describe in the manuscript did not show or follow the behaviors of the small semiconductor lasers. Authors might also illustrate if the conventional physical descriptions are not enough for the nanocavity.

-- To verify the coupling between the GaN emission and the resonant mode of the photonic crystal nanobeam cavity, authors should show the PL spectrum aligned with the whole lasing spectrum.

Our response:

We thank the reviewer for expressing his/her interest in our results and for highlighting the technological work and theoretical description. We hope that, in the following response, we will be able to satisfactorily answer the additional questions raised by the reviewer.

The central question concerns the role of linewidth narrowing, and in particular the role of the modified Schawlow-Townes formula, in the identification of lasing in 'conventional' semiconductor-based devices. What we mean here by conventional is a semiconductor laser (e.g. a VCSEL or a ridge waveguide laser), where the spontaneous emission factor β is very small ($\beta \ll 1$), typically of the order of 10^{-5} or less. For such devices, with output powers in the mW range, linewidth narrowing below and above threshold is pronounced and thus rightfully an established tool for verifying lasing. When it comes to nanolasers with a high β -factor, with output powers typically well below $1 \mu\text{W}$, the usual signatures of lasing (including linewidth narrowing) are far less pronounced and regularly deviate from the behavior of conventional lasers (i.e. Schawlow-Townes), see e.g. experimental Refs. [2-3], [20], [22-23], [27] from the main manuscript, additionally e.g. [Tatebayashi et al. Nat. Phot. **9**, 501-505 (2015)], [Kim et al. Nano Lett. **17**, 3465-3470 (2017)], [Hayenga et al. Optica **3**, 1187-1193 (2016)], as well as the theoretical description in [26]. In particular, most works show a constant or increasing linewidth above the threshold nonlinearity. The deviation from the conventional linewidth behavior is mainly attributed to the small number of photons ($\ll 1000$), cf. e.g. Refs. [6] & [32], in the lasing mode (and the correspondingly limited coherence) and additional broadening through refractive index fluctuations (gain-refractive index coupling) in the threshold region [26]. Consequently, the linewidth of a high- β nanolaser can stay constant in a large range of excitation powers above threshold, as discussed in e.g. [26], and additionally be subject to broadening mechanisms (e.g. of thermal origin) – see also answer to point 2) below.

The lack of unambiguous lasing signatures in high- β nanolasers was our main motivation to investigate the photon statistics through second-order autocorrelation measurements. The strength of this method is that it gives direct access to the underlying light generation/amplification mechanism of the emitted radiation and, thereby, allows one to prove or rebut lasing *while being immune to effects such as linewidth broadening*. Interestingly, as shown in a recent paper by our group [32] and prominently highlighted in [Hayenga, W. E., Khajavikhan, M., News and Views, *Light: Science & Applications* **6**, (2017)], even a non-linear I-O dependence in conjunction with a pronounced power dependent linewidth narrowing can be wrongly interpreted as a proof of lasing as revealed by $g^{(2)}(0)$ measurements (see micropillar C in Ref. [32]). To this end, the power dependent transition from a thermal to a Poissonian statistic is the most fundamental signature of the underlying change in the light amplification mechanism from spontaneous to predominantly stimulated emission and an analysis of the photon statistics through second-order autocorrelation is the most reliable proof of lasing in high- β micro- and nanolasers.

Action taken:

In order to further clarify the role and limits of conventional lasing indicators for high- β nanolasers, we have amended the corresponding section in the manuscript further. A new section on the investigation of heating in the cavity region has been added to the SM - see direct answer to point 2).

pages 5-6:

It is important to note that soft nonlinearities in the I-O curve, similar to that observed in Fig. 2a, can in principle also be related to trap filling [30] and thus cannot give an unambiguous proof of lasing. In fact, the observation of further indications of stimulated emission is required to prove lasing, in particular towards the limit of a “thresholdless” laser with a linear I-O curve. Another conventional signature of the onset of lasing is a decrease in the emission linewidth at half maximum (FWHM) and an associated increase in temporal coherence at the transition from predominantly spontaneous emission to stimulated emission. In this context, a power dependent linewidth narrowing can also be caused by quenching of absorption losses, which complicates the correct interpretation of this lasing indicator in high- β lasers. In contrast to conventional lasers (for which $\beta \ll 1$) like e.g. vertical cavity surface emitting or ridge waveguide lasers, where a pronounced linewidth reduction is an established lasing criterion, high- β nanolasers typically show only minor changes in linewidth at threshold [2-3], [23], [27] and usually deviate from the modified Schawlow-Townes formula for semiconductor based lasers [31]. Through the soft onset of lasing, which takes place at low carrier densities and with few intracavity photons [6], refractive index fluctuations (gain-refractive index coupling) persist over a large range of excitation powers around threshold and can lead to an almost constant linewidth in the threshold region and beyond [26]. Depending on the heat transport properties of the underlying design, heating of the cavity can also impact the linewidth under high excitation. Considering the discussion above, a lasing threshold could be falsely identified from these classical indicators [32].

In the present case, we observe a minor decrease in the emission linewidth at $P \approx 5 \text{ kW cm}^{-2}$, superimposed on an overall increasing linewidth (cf. Fig. 2b). Linewidth and emission wavelength of the reference cavity (cf. Fig. 2e) show a similar trend, although a decreasing linewidth at intermediate excitation power densities is not observed. The simultaneous start of the resonance redshift and the increase in linewidth suggest a thermal origin. Heating of the freestanding nanobeam membrane occurs at higher excitation powers, the resulting thermal expansion of the nanobeam leading to a redshift of the resonance wavelength with increasing excitation power density (cf. Figs. 2b, e). Temperature induced fluctuations in the cavity mode position on timescales faster than the minimum integration time accessible in experiment (10 ms) can result in additional broadening in photoluminescence due to spectral jitter. Similar observations to our experiment were made in Refs. [20] and [22]. Excitation power density dependent Raman thermometry measurements and accompanying thermal transport simulations confirm a temperature increase in the cavity region by more than 50 K in the high excitation range during our room temperature experiments (cf. Supplementary Figs. S4 and S5 and corresponding section in the SM). The thermal properties of the nanobeams could be improved, however at the cost of reducing Q and β , by coupling the cavity region directly to the substrate, using

e.g. the design adopted in Ref. [12], where a nanopillar under the cavity is used for electrical injection. As the conventionally sought-after lasing signatures (a pronounced I-O nonlinearity and linewidth decrease) are far more elusive, unambiguous proof of the onset of stimulated emission in high- β nanolasers requires excitation power dependent second-order autocorrelation measurements in order to monitor the change in emission statistics.

Reviewer:

2) The LL turn-on curve from the photonic crystal cavity did not show the obvious roll-over behavior due to the thermal effects under the high power pumping. Why authors attribute the unusual linewidth increase of the spectrum linewidth to the heating of the cavity under higher power pumping?

Our response:

The simultaneous start of a redshift in the emission wavelength and linewidth broadening indicate a common origin. A thermal expansion of the nanobeam along a temperature gradient from the cavity center explains a redshift of the cavity resonance with an associated temperature induced line broadening. The temperature dependent structural changes do not necessarily have to coincide with a thermal rollover, which we would expect for the given material system for temperatures of the active region approaching 100°C, e.g. [M. Kneissl et al. *Appl. Phys. Lett.* **75**, 581-583 (1999)]. To investigate the heating of the cavity region we have conducted Raman thermometry measurements as well as thermal transport simulations (using COMSOL Multiphysics). Both indicate (in good agreement) a temperature increase of 50-80 K in the cavity region in the high excitation regime. These results were previously not included in the manuscript and are now part of the supplementary material. Additional influences of line fluctuations on timescales below the minimum integration time (10 ms) of the spectra cannot be separated from the heating effects described above and are probably also present.

Action taken:

In addition to the amended text (direct answer above), a section on the investigation of heating in the nanobeam cavity has been added to the SM:

SM pages 9-11:

C. Investigation of sample heating

Heating of the nanobeam cavity was investigated by means of Raman thermometry and thermal transport simulations using COMSOL Multiphysics. For the Raman thermometry measurements the sample was excited under the same conditions as during the room temperature lasing characterization (methods section), using a 325 nm ultrasteep dielectric edge filter. Spectra were calibrated using a mercury gas discharge lamp and the spectrometer was not moved between measurements. The temperature increase in the cavity region was then inferred from the characteristic redshift of the polar $A_1(LO)$ phonon peak of the GaN matrix material [7]. We chose the $A_1(LO)$ phonon as a temperature sensor, as it is the most prominent Raman mode under the given resonant pumping conditions. An influence of the induced carrier density on the $A_1(LO)$ phonon (which would manifest itself in a blueshift of the resulting longitudinal phonon plasmon mode – the so-called LPP^+ mode) can be excluded [8-9]. Figure S4 shows the excitation power density dependent shift of the GaN $A_1(LO)$ mode, as well as the shift of the dominating first-order mode of silicon, indicating the temperature of the illuminated region of the substrate, for comparison. In linear approximation the Raman shift RS of the two peaks is given as $dT/dRS(\text{GaN } A_1(LO)) = -39 \text{ K per } cm^{-1}$ and $dT/dRS(\text{silicon}) = -46 \text{ K per } cm^{-1}$ near room temperature [7], [10]. From the shifts in Fig. S4, the temperature of the cavity region increases by $\sim 56 \text{ K}$, whereas the silicon substrate shows only a minor increase in temperature of $\sim 8 \text{ K}$.

Figure S4 | Raman thermometry measurements. Excitation power density dependent shift of the GaN $A_1(LO)$ and the first-order mode of silicon. From the overall redshift of the phonon modes, the temperature increase of cavity and substrate can be estimated to 56 K and 8 K, respectively.

Accompanying thermal transport simulations were carried out using COMSOL Multiphysics. For the simulations, a heat source was placed in the cavity region and the resulting temperature distribution along the nanobeam was subsequently simulated. Figure S5 shows the temperature distribution (shown for half the nanobeam) for several input powers. Solid and dashed lines correspond to a nanobeam with and without holes, respectively, indicating the lowered effective thermal conductivity in the actual structure. We can estimate an input power of $\sim 200\text{-}300\ \mu\text{W}$ (taking into account excitation density, illuminated cavity area and absorption coefficient) in the high excitation range during the characterization of the nanobeams, which results in an increase in cavity temperature of about 50-80 K in good agreement with the Raman thermometry. Note that the Raman measurements naturally involve an averaging over the laser spotsize, which comprises a gradient for the excitation power density.

Figure S5 | Thermal transport simulations. Temperature distribution along one half of the nanobeam (symmetric) for several input powers. Solid and dashed lines represent a nanobeam with and without holes, respectively. Inset: Dependence of the peak temperature on the input power.

Reviewer:

3) I also notice that there is only one InGaN/GaN QW in the photonic crystal laser cavity which is quite different to most of photonic crystal lasers. However, to support enough gain within such small cavity and overcome the optical loss within such compact cavities, 3-5 pairs of quantum wells embedded in the photonic crystal cavity is the most common vertical structure. What are the advantages for this 1 pair QW in the GaN nanobeam laser cavity?

Our response:

A single QW was chosen for sake of simplicity. Increasing the number of QWs would also increase the difficulty in maintaining an excellent crystal film quality during growth and would increase the absorption losses below threshold. Besides of that, adding more QWs would obviously increase the gain, but we would like to note that the material gain in III-nitrides is already larger than in more conventional III-V compound semiconductors. Transparency and required threshold carrier density have been estimated (cf. supplementary material of [13]) and are consistent with lasing under the given excitation.

REVIEWERS' COMMENTS:

Reviewer #4 (Remarks to the Author):

The revised manuscript, titled "On thresholdless lasing features in high- β nitride nanobeam cavities: a quantum optical study", by Stefan T. Jagsch et al. reports a high- β GaN-based photonic crystal nanobeam cavity laser with the experimental data and theoretical analysis. In the current revised manuscript and all supporting details, the authors had made strong efforts and aggressive responses to defend the works. Frankly speaking, this is a very nice work with a lot of theoretical details and modelling. However, the obvious lasing data and spectral details of the GaN photonic crystal laser are still missing. Now it is up to the editor to decide if the work could merit publication in Nature Communications.